# Surrogate NAS Benchmarks: Going Beyond the Limited Search Spaces of Tabular NAS Benchmarks

**Arber Zela**[1]*, **Julien Siems**[1]*, **Lukas Zimmer**[1]*, **Jovita Lukasik**[2],
**Margret Keuper**[2], **Frank Hutter**[1,3]
[1] University of Freiburg, [2] University of Mannheim, [3] Bosch Center for AI

## Abstract

The most significant barrier to the advancement of Neural Architecture Search (NAS) is its demand for large computational resources, which hinders scientifically sound empirical evaluations of NAS methods. Tabular NAS benchmarks have alleviated this problem substantially, making it possible to properly evaluate NAS methods in seconds on commodity machines. However, an unintended consequence of tabular NAS benchmarks has been a focus on extremely small architectural search spaces since their construction relies on exhaustive evaluations of the space. This leads to unrealistic results that do not transfer to larger spaces. To overcome this fundamental limitation, we propose a methodology to create cheap NAS surrogate benchmarks for arbitrary search spaces. We exemplify this approach by creating surrogate NAS benchmarks on the existing tabular NAS-Bench-101 and on two widely used NAS search spaces with up to $10^{21}$ architectures ($10^{13}$ times larger than any previous tabular NAS benchmark). We show that surrogate NAS benchmarks can model the true performance of architectures better than tabular benchmarks (at a small fraction of the cost), that they lead to faithful estimates of how well different NAS methods work on the original non-surrogate benchmark, and that they can generate new scientific insight. We open-source all our code and believe that surrogate NAS benchmarks are an indispensable tool to extend scientifically sound work on NAS to large and exciting search spaces.

## 1 Introduction

Neural Architecture Search (NAS) has seen huge advances in search efficiency, but the field has recently been criticized substantially for non-reproducible research, strong sensitivity of results to carefully-chosen training pipelines, hyperparameters and even random seeds (Yang et al., 2020; Li & Talwalkar, 2020; Lindauer & Hutter, 2020; Shu et al., 2020; Yu et al., 2020). A leading cause that complicates reproducible research in NAS is the computational cost of even just single evaluations of NAS algorithms, not least in terms of carbon emissions (Patterson et al., 2021; Li et al., 2021a).

Tabular NAS benchmarks, such as NAS-Bench-101 (Ying et al., 2019) and NAS-Bench-201 (Dong & Yang, 2020), have been a game-changer for reproducible NAS research, for the first time allowing scientifically sound empirical evaluations of NAS methods with many seeds in minutes, while ruling out confounding factors, such as different search spaces, training pipelines or hardware/software versions. This success has recently led to the creation of many additional tabular NAS benchmarks, such as NAS-Bench-1shot1 (Zela et al., 2020b), NATS-Bench (Dong et al., 2021), NAS-HPO-bench (Klein & Hutter, 2019), NAS-Bench-NLP (Klyuchnikov et al., 2020), NAS-Bench-ASR (Mehrotra et al., 2021), and HW-NAS-Bench (Li et al., 2021b) (see Appendix A.1 for more details on these previous NAS benchmarks).

However, these tabular NAS benchmarks rely on an exhaustive evaluation of *all* architectures in a search space, limiting them to unrealistically small search spaces (so far containing only between 6k and 423k architectures). This is a far shot from standard spaces used in the NAS literature, which

---

*Equal contribution. Email to: `{zelaa}@cs.uni-freiburg.de`

contain more than $10^{18}$ architectures (Zoph & Le, 2017; Liu et al., 2019; Wu et al., 2019a). This discrepancy can cause results gained on tabular NAS benchmarks to not generalize to realistic search spaces; e.g., promising anytime results of local search on tabular NAS benchmarks were indeed shown to not transfer to realistic search spaces (White et al., 2020a).

Making things worse, as discussed in the panel of the most recent NAS workshop at ICLR 2021, to succeed in automatically discovering qualitatively new types of architectures (such as, e.g., Transformers (Vaswani et al., 2017)) the NAS community will have to focus on even more expressive search spaces in the future. To not give up the recent progress in terms of reproducibility that tabular NAS benchmarks have brought, we thus need to develop their equivalent for arbitrary search spaces. That is the goal of this paper.

**Our contributions.** Our main contribution is to introduce the concept of *surrogate NAS benchmarks* that can be constructed for arbitrary NAS search spaces and allow for the same cheap query interface as tabular NAS benchmarks. We substantiate this contribution as follows:

1. We demonstrate that a surrogate fitted on a subset of architectures can model the true performance of architectures *better* than a tabular benchmark (Section 2).

2. We showcase our methodology by building surrogate benchmarks on a realistically-sized NAS search space (up to $10^{21}$ possible architectures, i.e., $10^{13}$ times more than any previous tabular NAS benchmark), thoroughly evaluating a range of regression models as surrogate candidates, and showing that strong generalization performance is possible even in large spaces (Section 3).

3. We show that the search trajectories of various NAS optimizers running on the surrogate benchmarks closely resemble the ground truth trajectories. This enables sound simulations of runs usually requiring thousands of GPU hours in a few seconds on a single CPU machine (Section 3).

4. We demonstrate that surrogate benchmarks can help in generating new scientific insights by rectifying a previous hypothesis on the performance of local search in large spaces (Section 4).

To foster reproducibility, we open-source all our code, data, and surrogate NAS benchmarks.[1]

## 2 MOTIVATION – CAN WE DO BETTER THAN A TABULAR BENCHMARK?

We start by motivating the use of surrogate benchmarks by exposing an issue of tabular benchmarks that has largely gone unnoticed. Tabular benchmarks are built around a costly, exhaustive evaluation of *all* possible architectures in a search space, and when an architecture's performance is queried, the tabular benchmark simply returns the respective table entry. The issue with this process is that the stochasticity of mini-batch training is also reflected in the performance of an architecture $i$, hence making it a random variable $Y_i$. Therefore, the table only contains results of a few draws $y_i \sim Y_i$ (existing NAS benchmarks feature up to 3 runs per architecture). Given the variance in these evaluations, a tabular benchmark acts as a simple estimator that assumes *independent*

| Model | Mean Absolute Error (MAE) | | |
|---|---|---|---|
| | 1, [2, 3] | 2, [1, 3] | 3, [1, 2] |
| Tab. | 4.534$e$-3 | 4.546$e$-3 | 4.539$e$-3 |
| Surr. | **3.446e-3** | **3.455e-3** | **3.441e-3** |

Table 1: MAE between performance predicted by a tab./surr. benchmark fitted with one seed each, and the true performance of evaluations with the two other seeds. Test seeds in brackets.

random variables, and thus estimates the performance of an architecture based only on previous evaluations of the same architecture. From a machine learning perspective, knowing that similar architectures tend to yield similar performance and that the variance of individual evaluations can be high (both shown to be the case by Ying et al. (2019)), it is natural to assume that better estimators may exist. In the remainder of this section, we empirically verify this hypothesis and show that surrogate benchmarks can provide *better* performance estimates than tabular benchmarks based on *less* data.

**Setup.** For the analysis in this section, we choose NAS-Bench-101 (Ying et al., 2019) as a tabular benchmark and a Graph Isomorphism Network (GIN, Xu et al. (2019a)) as our surrogate model. [2] Each architecture $x_i$ in NAS-Bench-101 contains 3 validation accuracies $y_i^1, y_i^2, y_i^3$ from training $x_i$

---

[1]https://github.com/automl/nasbench301
[2]We used a GIN implementation by Errica et al. (2020); see Appendix B for details on training the GIN.

with 3 different seeds. We excluded all diverged models with less than 50% validation accuracy on any of the three evaluations in NAS-Bench-101. We split this dataset to train the GIN surrogate model on one of the seeds, e.g., $\mathcal{D}^{train} = \{(x_i, y_i^1)\}_i$ and evaluate on the other two, e.g., $\mathcal{D}^{test} = \{(x_i, \bar{y}_i^{23})\}_i$, where $\bar{y}_i^{23} = (y_i^2 + y_i^3)/2$.

**Results.** We compute the mean absolute error MAE $= \frac{\sum_i |\hat{y}_i - \bar{y}_i^{23}|}{n}$ of the surrogate model trained on $\mathcal{D}^{train} = \{(x_i, y_i^1)\}_i$, where $\hat{y}_i$ is the predicted validation accuracy and $n = |\mathcal{D}^{test}|$. Table 1 shows that the surrogate model yields a lower MAE than the tabular benchmark, i.e. MAE $= \frac{\sum_i |y_i^1 - \bar{y}_i^{23}|}{n}$. We also report the mean squared error and Kendall tau correlation coefficient in Table 4 in the appendix showing that the ranking between architectures is also predicted better by the surrogate. We repeat the experiment in a cross-validation fashion w.r.t to the seeds and conclude: *In contrast to a single tabular entry, the surrogate model learns to smooth out the noise.*[3]

Next, we fit the GIN surrogate on subsets of $\mathcal{D}^{train}$ and show in Figure 1 how its performance scales with the amount of training data used. The surrogate model performs better than the tabular benchmark when the training set has more than $\sim$21,500 architectures. (Note that $\mathcal{D}^{test}$ remains the same as in the previous experiment, i.e., it includes all 423k architectures in NAS-Bench-101.) As a result, we conclude that: *A surrogate model can yield strong predictive performance when only a subset of the search space is available as training data.*

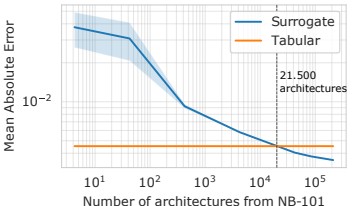

Figure 1: Number of architectures used for training the GIN surrogate model vs MAE on the NAS-Bench-101 dataset.

These empirical findings suggest that we can create reliable surrogate benchmarks for much larger and more realistic NAS spaces, which are infeasible to be exhaustively evaluated (as would be required to construct tabular benchmarks).

## 3 GOING BEYOND SPACE SIZE LIMITS WITH SURROGATE BENCHMARKS

We now introduce the general methodology that we propose to effectively build realistic and reliable surrogate NAS benchmarks. We showcase this methodology by building surrogate benchmarks on two widely used search spaces and datasets, namely DARTS (Liu et al., 2019) + CIFAR-10 and FBNet (Wu et al., 2019a) + CIFAR-100. Having verified our surrogate NAS benchmark methodology on these well-known spaces, we strongly encourage the creation of additional future surrogate NAS benchmarks on a broad range of large and exciting search spaces, and the compute time saved by replacing expensive real experiments on the DARTS or FBNet space with cheap experiments on our surrogate versions of them might already be used for this purpose.

We name our surrogate benchmarks depending on the space and surrogate model considered as follows: `Surr-NAS-Bench-{space}-{surrogate}` (or, `SNB-{space}-{surrogate}` for short). For example, we introduce `SNB-DARTS-GIN`, which is a surrogate benchmark on the DARTS space and uses a GIN (Xu et al., 2019a) as a surrogate model.

### 3.1 GENERAL METHODOLOGY AND SURR-NAS-BENCH-DARTS

We first explain our methodology for the DARTS (Liu et al., 2019) search space, which consists of more than $10^{18}$ architectures. We selected this search space for two main reasons: (1) due to the huge number of papers building on DARTS and extending it, the DARTS search space, applied to CIFAR-10 is the most widely used non-tabular NAS benchmark in the literature, and as such it provides a convincing testbed for our surrogate benchmarks[4]; and (2) the surrogate benchmark we construct frees up the substantial compute resources that are currently being invested for experiments

---

[3]We note that the average estimation error of tabular benchmarks could be reduced by a factor of $\sqrt{k}$ by performing $k$ runs per architecture. The error of surrogate models would also shrink when they are based on more data, but as $k$ grows large tabular benchmarks would become competitive with surrogate models.

[4]In particular, the alternative of first constructing a new non-tabular benchmark and then building a surrogate benchmark for it would have been susceptible to many confounding factors.

on this non-tabular benchmark; we hope that these will instead be used to study additional novel and more exciting spaces and/or datasets.

### 3.1.1 SURROGATE MODEL CANDIDATES

We are interested in predicting a metric $y \in Y \subseteq \mathbb{R}$, such as accuracy or runtime, given an architecture encoding $x \in X$. A *surrogate* model can be any regression model $\phi : X \to Y$, however, picking the right surrogate to learn this mapping can be non-trivial. This of course depends on the structure and complexity of the space, e.g., if the architecture blocks have a hierarchical structure or if they are just composed by stacking layers sequentially, and the number of decisions that have to be made at every node or edge in the graph.

Due to the graph representation of the architectures commonly used in NAS, Graph Convolutional Networks (GCNs) are frequently used as NAS predictors (Wen et al., 2019; Ning et al., 2020; Lukasik et al., 2021). In particular, the GIN (Xu et al., 2019a) is a good fit since several works have found it to perform well on many benchmark datasets (Errica et al., 2020; Hu et al., 2020; Dwivedi et al., 2020), especially when the space contains many isomorphic graphs. Other interesting choices could be the *DiffPool* (Ying et al., 2018) model, which can effectively model hierarchically-structured spaces, or Graph Transformer Networks (GTN) (Yun et al., 2019) for more complex heterogeneous graph structures.

Nevertheless, as shown in White et al. (2021b), simpler models can already provide reliable performance estimates when carefully tuned. We compare the GIN to a variety of common regression models. We evaluate Random Forests (RF) and Support Vector Regression (SVR) using implementations from scikit-learn (Pedregosa et al., 2011). We also evaluate the tree-based gradient boosting methods XGBoost (Chen & Guestrin, 2016), LGBoost (Ke et al., 2017) and NGBoost (Duan et al., 2020), recently used for predictor-based NAS (Luo et al., 2020). We comprehensively review architecture performance prediction in Appendix A.2.

We would like to note the relation between our surrogate model candidates and performance predictors in a NAS algorithm (White et al., 2021b), e.g., in Bayesian Optimization (BO). In principle, any type of NAS predictor can be used, including zero-shot proxies (Mellor et al., 2021; Abdelfattah et al., 2021), one-shot models (Brock et al., 2018; Bender et al., 2018; 2020; Zhao et al., 2021), learning curve extrapolation methods (Domhan et al., 2015; Baker et al., 2017; Klein et al., 2017), and model-based proxies (Ru et al., 2021; Ma et al., 2019; Shi et al., 2020; White et al., 2021a). Since White et al. (2021b) found model-based proxies to work best in the regime of relatively high available initialization time (the time required to produce the training data) and low query time (the time required to make a prediction) we use these types of performance predictors for our surrogates. If we were also willing to accept somewhat higher query times, then combined models that extrapolate initial learning curves based on full learning curves of previous architectures (Baker et al., 2017; Klein et al., 2017) would be a competitive alternative (White et al., 2021b).

### 3.1.2 DATA COLLECTION

As the search spaces we aim to attack with surrogate NAS benchmarks are far too large to be exhaustively evaluated, care has to be taken when sampling the architectures which will be used to train the surrogate models. Sampling should yield good overall coverage of the architecture space while also providing a special focus on the well-performing regions that optimizers tend to exploit.

Our principal methodology for sampling in the search space is inspired by Eggensperger et al. (2015), who collected unbiased data about hyperparameter spaces by random search (RS), as well as biased and denser samples in high-performance regions by running hyperparameter optimizers. This is desirable for a surrogate benchmark since we are interested in evaluating NAS methods that exploit such good regions of the space.

We now describe how we collected the dataset `SNB-DARTS` we use for our case study of surrogate NAS benchmarks on the DARTS search space. The search space itself is detailed in Appendix C.1. Table 5 in the appendix lists the 10 NAS methods we used to collect architectures in good regions of the space, and how many samples we collected with each (about 50k in total). Additionally, we evaluated ∼1k architectures in poorly-performing regions for better coverage and another ∼10k for the analysis conducted on the dataset and surrogates. We refer to Appendices C.2 and C.3 for

details on the data collection and the optimizers, respectively. Appendix C.4 shows the performance of the various optimizers in this search space and visualizes their overall coverage of the space, as well as the similarity between sampled architectures using t-SNE (van der Maaten & Hinton, 2008) in Figure 12. Besides showing good overall coverage, some well-performing architectures in the search space form distinct clusters which are mostly located outside the main cloud of points. This clearly indicates that architectures with similar performance are close to each other in the architecture space. We also observe that different optimizers sample different architectures (see Figure 13 in the appendix). Appendix C.5 provides statistics of the space concerning the influence of the cell topologies and the operations, and Appendix C.6 describes the full training pipeline used for all architectures. In total, our `SNB-DARTS` dataset consists of ~60k architectures and their performances on CIFAR-10 (Krizhevsky, 2009). We split the collected data into train/val/test splits, which we will use to train, tune and evaluate our surrogates throughout the experiments.

Finally, we would like to point out that, in hindsight, adding training data of well-performing regions may be less important for a surrogate NAS benchmark than for a surrogate HPO benchmark: Appendix F.3 shows that surrogates based purely on random evaluations also yield competitive performance. We believe that this is a result of HPO search spaces containing many configurations that yield dysfunctional models, which is less common for architectures in many NAS search spaces, hence allowing random search to cover the search space well enough to build strong surrogates.

### 3.1.3 EVALUATING THE DATA FIT

Similarly to Wen et al. (2019), Baker et al. (2017) and White et al. (2021b), we assess the quality of the data fit via the coefficient of determination ($R^2$) and the Kendall rank correlation coefficient ($\tau$). Since Kendall $\tau$ is sensitive to noisy evaluations, following Yu et al. (2020) we use a sparse Kendall Tau (sKT), which ignores rank changes at 0.1% accuracy precision, by rounding the predicted validation accuracy prior to computing $\tau$. We compute such metrics on a separate test set of architectures, never used to train or tune the hyperparameters of the surrogate models.

As we briefly mentioned above, our main goal is to obtain similar learning trajectories (architecture performance vs. runtime) when running NAS optimizers on the surrogate benchmark and the real benchmark. The ranking of architectures is therefore one of the most important metrics to pay attention to since most NAS optimizers are scale-invariant, i.e., they will find the same solution for the function $f(x)$ and $a \cdot f(x)$, with scalar $a$.

For applying the surrogate model candidates described above on our dataset `SNB-DARTS`, we tuned the hyperparameters of all surrogate models using the multi-fidelity Bayesian optimization method BOHB (Falkner et al., 2018); details on their respective hyperparameter search spaces are given in Table 6 in the appendix. We use train/val/test splits (0.8/0.1/0.1) stratified across the NAS methods used for the data collection. This means that the ratio of architectures from a particular optimizer is constant across the splits, e.g., the test set contains 50% of its architectures from RS since RS was used to obtain 50% of the total architectures we trained and evaluated. We provide additional details on the preprocessing of the architectures for the surrogate models in Appendix E.1. As Table 2 shows, our three best-performing models were LGBoost, XGBoost, and GIN; therefore, we focus our analysis on these in the following.

| Model | Test | |
|---|---|---|
| | $R^2$ | sKT |
| LGBoost | **0.892** | 0.816 |
| XGBoost | 0.832 | **0.817** |
| GIN | 0.832 | 0.778 |
| NGBoost | 0.810 | 0.759 |
| $\mu$-SVR | 0.709 | 0.677 |
| MLP (Path enc.) | 0.704 | 0.697 |
| RF | 0.679 | 0.683 |
| $\epsilon$-SVR | 0.675 | 0.660 |

Table 2: Performance of different regression models fitted on the `SNB-DARTS` dataset.

In addition to evaluating the data fit on our data splits, we investigate the impact of parameter-free operations and the cell topology in Appendices E.6 and E.7, respectively. We find that all of LGBoost, XGBoost, and GIN accurately predict the drop in performance when increasingly replacing operations with parameter-free operations in a normal cell of the DARTS search space.

### 3.1.4 MODELS OF RUNTIME AND OTHER METRICS

To allow evaluations of multi-objective NAS methods, and to allow using "simulated wallclock time" on the x axis of plots, we also predict the runtime of architecture evaluations. For this, we train an LGB model with the runtime as targets (see Appendix E.4 for details); this runtime is also logged for all architectures in our dataset `SNB-DARTS`. Runtime prediction is less challenging than performance

prediction, resulting in an excellent fit of our LGB runtime model on the test set (sKT: 0.936, $R^2$: 0.987). Other metrics of architectures, such as the number of parameters and multiply-adds, do not require a surrogate model but can be queried exactly.

### 3.1.5 NOISE MODELING

The aleatoric uncertainty in the architecture evaluations is of practical relevance since it not only determines the robustness of a particular architecture when trained with a stochastic optimization algorithm, but it can also steer the trajectories of certain NAS optimizers and yield very different results when run multiple times. Therefore, modeling such noise in the architecture evaluations is an important step towards proper surrogate benchmarking.

A simple way to model such uncertainty is to use ensembles of the surrogate model. Ensemble methods are commonly used to improve predictive performance (Dietterich, 2000). Moreover, ensembles of deep neural networks, so-called deep ensembles (Lakshminarayanan et al., 2017), have been proposed as a simple and yet powerful (Ovadia et al., 2019) way to obtain predictive uncertainty. We therefore create an ensemble of 10 base learners for each of our three best performing models (GIN, XGB, LGB) using a 10-fold cross-validation for our train and validation split, as well as different initializations.

| Model | MAE 1, [2,3,4,5] | Mean $\sigma$ | KL div. |
|---|---|---|---|
| Tabular | 1.38e−3 | undef. | undef. |
| GIN | **1.13e-3** | 0.6e−3 | **16.4** |
| LGB | 1.33e−3 | 0.3e−3 | 68.9 |
| XGB | 1.51e−3 | 0.3e−3 | 134.4 |

Table 3: Metrics for the selected surrogate models on 500 architectures that were evaluated 5 times.

To assess the quality of our surrogates' predictive uncertainty, we compare the predictive distribution of our ensembles to the ground truth. We assume that the noise in the architecture performance is normally distributed and compute the Kullback–Leibler (KL) divergence between the ground truth accuracy distribution and predicted distribution.

### 3.1.6 PERFORMANCE OF SURROGATES VS. TABLE LOOKUPS

We now mirror the analysis we carried out for our GIN surrogate on the `NB-101` dataset in our motivational example (Section 2), but now using our higher-dimensional dataset `SNB-DARTS`. For this, we use a set of 500 architectures trained with 5 seeds each. We train the surrogates using only one evaluation per architecture (i.e., seed 1) and take the mean accuracy of the remaining ones as ground truth value (i.e., seeds 2-5). We then compare against a tabular model with just one evaluation (seed 1). Table 3 shows that, as in the motivational example, our GIN and LGB surrogate models yield estimates closer to the ground truth than the table lookup based on one evaluation. This confirms our main finding from Section 2, but this time on a much larger search space. In terms of noise modeling, we find the GIN ensemble to quite clearly provide the best estimate.

### 3.1.7 EVALUATING THE NAS SURROGATE BENCHMARKS `SNB-DARTS-XGB` AND `SNB-DARTS-GIN`

Having assessed the ability of the surrogate models to fit the search space, we now evaluate the surrogate NAS benchmarks `SNB-DARTS-XGB` and `SNB-DARTS-GIN`, comparing cheap evaluations of various NAS algorithms on them against their expensive counterpart on the original non-tabular NAS benchmarks. We note that for the surrogate trajectories, *each architecture evaluation is sampled from the surrogate model's predictive distribution for the given architecture*. Therefore, different optimizer runs lead to different trajectories.

We first compare the trajectories of blackbox optimizers on the true, non-tabular benchmark vs. on the surrogate benchmark, using surrogates trained on all data. For the true benchmark, we show the trajectories we ran to create our dataset (based on a single run per optimizer, since we could not afford repetitions due to the extreme compute requirements of 115 GPU days for a single run). For the evaluations on the surrogate, on the other hand, we can trivially afford to perform multiple runs.

**Results (all data).** As Figure 2 shows, both the XGB and the GIN surrogate capture behaviors present in the true benchmark. For instance, the strong improvements of BANANAS and RE are also present on the surrogate benchmark at the correct time. In general, the ranking of the optimizers towards convergence is accurately reflected on the surrogate benchmark. Also, the initial random exploration of algorithms like TPE, RE and DE is captured as the large initial variation in performance

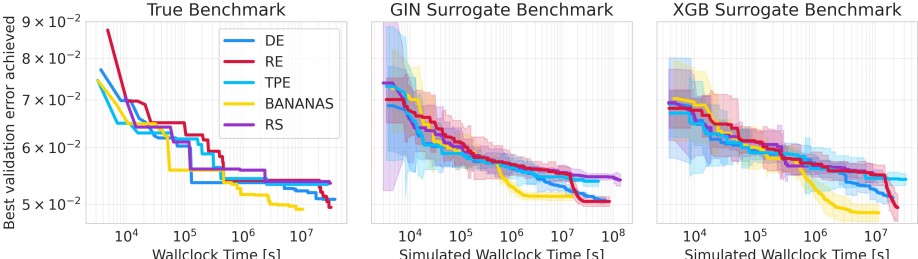

Figure 2: Anytime performance of different optimizers on the real benchmark (left) and the surrogate benchmark (GIN (middle) and XGB (right)) when training ensembles on data collected from all optimizers. Trajectories on the surrogate benchmark are averaged over 5 optimizer runs and the standard deviation is depicted.

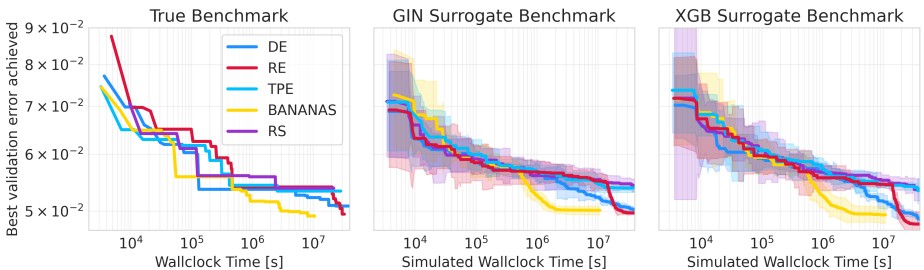

Figure 3: Anytime performance of different optimizers on the real benchmark (left) and the surrogate benchmark (GIN (middle) and XGB (right)) when training ensembles only on data collected by random search. Trajectories on the surrogate benchmark are averaged over 5 optimizer runs.

indicates. Notably, the XGB surrogate ensemble exhibits a high variation in well-performing regions as well and seems to slightly underestimate the error of the best architectures. The GIN surrogate, on the other hand, shows less variance in these regions but slightly overpredicts for the best architectures.

Note, that due to the size of the search space, random search stagnates and cannot identify one of the best architectures even after tens of thousands of evaluations, with BANANAS finding better architectures orders of magnitude faster. This is correctly reflected in both surrogate benchmarks.

Finally, surrogate NAS benchmarks can also be used to monitor the behavior of one-shot NAS optimizers throughout their search phase, by querying the surrogate model with the currently most promising discrete architecture. We show this evaluation in Appendix F.1.

### 3.1.8 ABLATION STUDIES

**Fitting only on random architectures.** We also investigate whether it is possible to create surrogate NAS benchmarks only based on random architectures. To that end, we studied surrogate models based only on the 23746 architectures explored by random search and find that we can indeed obtain realistic trajectories (see Figure 3) but lose some predictive performance in the well-performing regions. For the full description of this experiment see Appendix F.3. Nevertheless, such benchmarks have the advantage of not possibly favouring any NAS optimizer used for the generation of training data. In an additional experiment, we found that surrogates built on only well-performing architectures (92% and above) yielded poor extrapolation to worse architectures, but that surrogate benchmarks based on them still yielded realistic trajectories. We attribute this to NAS optimizers' focus on good architectures. For details, see Appendix F.2.

**Leave One-Optimizer-Out (LOOO) analysis.** Similarly to Eggensperger et al. (2015), we also perform a leave-one-optimizer-out analysis (LOOO) analysis, a form of cross-validation on the optimizers we used for data collection, i.e., we assess each optimizer with surrogate benchmarks created based on data excluding that gathered by said optimizer (using a stratified 0.9/0.1 train/val split over the other NAS methods). Figure 16 in the appendix compares the trajectories obtained

from 5 runs in the LOOO setting on the surrogate benchmark to the groundtruth. The XGB and GIN surrogates again capture the general behavior of different optimizers well, illustrating that characteristics of new optimization algorithms can be captured with the surrogate benchmark.

## 3.2 USING THE GATHERED KNOWLEDGE TO CONSTRUCT SURR-NAS-BENCH-FBNET

In Section 3.1, we presented our general methodology for constructing surrogate NAS benchmarks and exemplified it on the DARTS space and CIFAR-10 dataset. In this section, we make use of the knowledge gathered from those experiments to construct a second surrogate benchmark on the very different but similarly popular FBNet (Wu et al., 2019a) search space and CIFAR-100 dataset.

The FBNet search space consists of a fixed macro architecture with 26 sequentially stacked layers in total. The first and last three layers are fixed, while the other in-between layers have to be searched. There are 9 operation choices for every such layer (3 kernel sizes, times 3 possible numbers of filter), yielding a total of $9^{22} \approx 10^{21}$ unique architecture configurations. See Appendix D for more details on the search space and training pipeline.

**Surrogate models choice.** Despite the high cardinality of the space, it has a less complex structure compared to the DARTS space. Because of this, and also based on its high predictive performance on the DARTS space, we pick XGBoost as a surrogate model for the FBNet space. Similarly to `SNB-DARTS`, we create an ensemble of 5 base learners trained with different initializations.

**Data collection and predictive performance.** The data we collected to create `SNB-FBNet` comprises 25k architectures sampled uniformly at random, split in train/val/test sets (0.8/0.1/0.1). Based on the performance of `SNB-DARTS-XGB` fitted only on randomly sampled data (see Appendix F.3), and in order to obtain a surrogate without sampling bias, we fit the XGB surrogate only on the randomly sampled architectures. Using the transferred hyperparameters from the tuned XGB surrogate on the DARTS space already yields 0.84 $R^2$ and 0.75 sKT correlation coefficient value on the test set. We additionally collected 1.6k, 1.5k, 1k, 500, 200 architectures sampled by HEBO (Cowen-Rivers et al., 2020), RE, TPE, DE and COMBO, respectively, which we will use to evaluate the `SNB-FBNet-XGB` benchmark below.

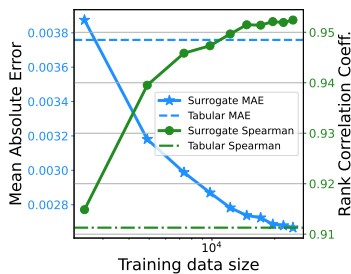

Figure 4: Number of architectures used for training the XGB surrogate model vs. MAE and rank correlation on the FBNet search space.

**Noise modeling and performance vs. table lookups.** In order to assess the performance of `SNB-FBNet-XGB` towards table lookups, we conduct a similar experiment as we did for `SNB-DARTS` in Section 3.1.6. We trained a set of 500 randomly sampled architectures 3 times each with different seeds. We then trained the surrogate model on only one seed and compare the predicted performance on this set of 500 architectures to the mean validation error of seeds 2 and 3. The latter serves as our "ground truth". Similarly, we compare our surrogate to a tabular model which contains only one evaluation per architecture, namely seed 1 in our case. Figure 4 shows that the XGB surrogate is already able to outperform the table entries in terms of mean absolute error towards the "ground truth" error when fitted using only 20% of the training data.

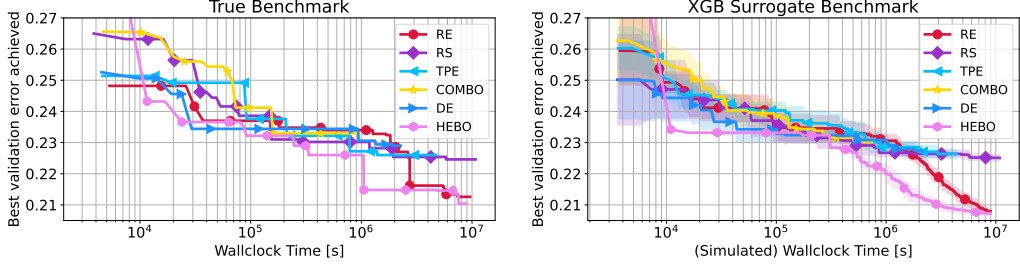

Figure 5: Anytime performance of different optimizers on the real benchmark and the XGB surrogate benchmark when training ensembles on data collected only from random search. Trajectories on the surrogate benchmark are averaged over 5 optimizer runs and the standard error is depicted.

**Evaluating the NAS surrogate benchmark `SNB-FBNet-XGB`.** In Figure 5, we show the black-box optimizer trajectories on `SNB-FBNet-XGB` (right plot) and the ones on the real FBNet benchmark (left plot). The XGB surrogate has low variance in well-performing regions of the space, with a slight overprediction of performance (compared to the single RE run possible on the original benchmark). Note that COMBO and TPE have higher overheads for fitting their internal models, which depend on the dimensionality of the architecture space being optimized. Our surrogate model successfully simulates both their performance and this overhead. Finally, similarly to the results observed in the DARTS space, random search cannot identify a good architecture even when evaluating 2500 architectures, with RE and HEBO being at least 8 and 16 times faster, respectively, in finding a better architecture. Also, in contrast to the DARTS space, here, the absolute difference between random search and HEBO and RE is larger, with 1-2% absolute difference in error.

## 4 USING NAS SURROGATE BENCHMARKS TO DRIVE NAS RESEARCH

Finally, we perform a case study that demonstrates how surrogate NAS benchmarks can drive NAS research. Coming up with research hypotheses and drawing conclusions when prototyping or evaluating NAS algorithms on less realistic benchmarks is difficult, particularly when these evaluations require high computational budgets. Surrogate NAS benchmarks alleviate this dilemma via their cheap and reliable estimates.

To showcase such a scenario, we evaluate Local Search[5] (LS) on our surrogate benchmarks `SNB-DARTS-GIN` and `SNB-DARTS-XGB`, as well as the original, non-tabular DARTS benchmark. White et al. (2020a) concluded that LS does not perform well on such a large space by running it for 11.8 GPU days ($\approx 10^6$ seconds), and we are able to reproduce the same results via `SNB-DARTS-GIN` and `SNB-DARTS-XGB` in a few seconds (see Figure 6). Furthermore, while White et al. (2020a) could not afford longer runs (nor repeats), on our surrogate NAS benchmarks this is trivial. Doing so suggests that LS shows qualitatively different behavior when run for an order of magnitude longer, transitioning from being the worst method (even worse than random search) to being one of the best. We verified this suggestion by running LS for longer on the actual DARTS benchmark (also see Figure 6). This allows us to revise the initial conclusion of White et al. (2020a) to: *The initial phase of LS works poorly for the DARTS search space, but in the higher-budget regime LS yields state-of-the-art performance.*

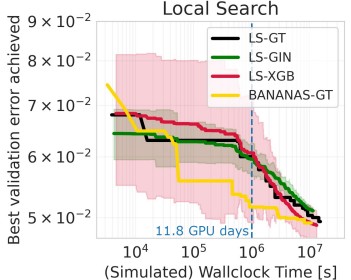

Figure 6: Case study results for Local Search. GT is the ground truth, GIN and XGB are results on `SNB-DARTS-GIN` and `SNB-DARTS-XGB`, respectively.

This case study shows how a surrogate benchmark was already used to cheaply obtain hints on a research hypothesis that lead to correcting a previous finding that only held for short runtimes. We look forward to additional uses of surrogate NAS benchmarks along such lines.

## 5 CONCLUSIONS AND FUTURE WORK

We proposed the concept of constructing cheap-to-evaluate surrogate NAS benchmarks that can be used as conveniently as tabular NAS benchmarks but in contrast to these can be built for search spaces of arbitrary size. We showcased this methodology by creating and evaluating several NAS surrogate benchmarks, including for two large search spaces with roughly $10^{18}$ and $10^{21}$ architectures, respectively, for which the exhaustive evaluation needed to construct a tabular NAS benchmark would be utterly infeasible. We showed that surrogate benchmarks can be more accurate than tabular benchmarks, assessed various possible surrogate models, and demonstrated that they can accurately simulate anytime performance trajectories of various NAS methods at a fraction of the true cost. Finally, we showed how surrogate NAS benchmarks can lead to new scientific findings.

In terms of future work, having access to a variety of challenging benchmarks is essential to the sustained development and evaluation of new NAS methods. We therefore encourage the community to expand the scope of current NAS benchmarks to different search spaces, datasets, and problem domains, utilizing surrogate NAS benchmarks to tackle exciting, larger and more complex spaces.

---

[5]We use the implementation and settings of Local Search provided by White et al. (2020a).

## 6 ETHICS STATEMENT

**Societal Impact.** We hope and expect that the NAS practitioner will benefit from our *surrogate* NAS benchmarks the same way for research on exciting large search spaces as she so far benefited from *tabular* NAS benchmarks for research on toy search spaces – by using them for fast and easy prototyping of algorithms and large-scale (yet cheap) scientific evaluations. We expect that in the next years this will save millions of GPU hours (which will otherwise be spent on running expensive NAS benchmarks) and thus help reduce the potentially hefty carbon emissions of NAS research (Hao, 2019). It will also ensure that the entire NAS community can partake in exploring large and exciting NAS design spaces, rather than only corporations with enormous compute power, thereby further democratizing NAS research.

**Limitations & Guidelines for Using Surrogate Benchmarks.** We would like to point out that there exists a risk that prior knowledge about the surrogate model used in a particular benchmark could lead to the design of algorithms that overfit that benchmark. To this end, we recommend the following best practices to ensure a safe and fair benchmarking of NAS methods on surrogate NAS benchmarks:

- The surrogate model should be treated as a black-box function, hence only be used for performance prediction and not exploited to extract, e.g., gradient information.
- We discourage benchmarking methods that internally use the same model as the surrogate model picked in the surrogate benchmark (e.g. GNN-based Bayesian optimization should not only be benchmarked using a GIN surrogate benchmark).
- We encourage running experiments on versions of surrogate benchmarks that are based on (1) all available training architectures and (2) only architectures collected with uninformed methods, such as random search or space-filling designs. As shown in Appendix F.3, (1) yields better predictive models, but (2) avoids any potential bias (in the sense of making more accurate predictions for architectures explored by a particular type of NAS optimizer) and can still yield strong benchmarks.
- In order to ensure comparability of results in different published papers, we ask users to state the benchmark's version number. So far, we release `SNB-DARTS-XGB-v1.0`, `SNB-DARTS-GIN-v1.0`, `SNB-DARTS-XGB-rand-v1.0`, `SNB-DARTS-GIN-rand-v1.0`, and `SNB-FBNet-XGB-rand-v1.0`.

Due to the flexibility of surrogate NAS benchmarks to tackle arbitrary search spaces, we expect that the community will create many such benchmarks in the future, just like it already created many tabular NAS benchmarks Ying et al. (2019); Dong & Yang (2020); Zela et al. (2020b); Dong et al. (2021); Klein & Hutter (2019); Klyuchnikov et al. (2020); Mehrotra et al. (2021); Li et al. (2021b). We therefore collect best practices for the creation of such new surrogate NAS benchmarks in Appendix G.

### ACKNOWLEDGMENTS

We thank the anonymous reviewers for suggesting very insightful experiments, in particular the experiments for NAS benchmarks based only on random architectures. The authors acknowledge funding by the Robert Bosch GmbH, by the European Research Council (ERC) under the European Union Horizon 2020 research and innovation programme through grant no. 716721, by BMBF grants DeToL and RenormalizedFlows (01IS19077C), and by the Deutsche Forschungsgemeinschaft (DFG, German Research Foundation) under grant number 417962828. This research was partially supported by TAILOR, a project funded by EU Horizon 2020 research and innovation programme under GA No 952215.

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

# A  RELATED WORK

## A.1  EXISTING NAS BENCHMARKS

Benchmarks for NAS were introduced only recently with NAS-Bench-101 (Ying et al., 2019) as the first among them. NAS-Bench-101 is a tabular benchmark consisting of ~423k unique architectures in a cell structured search space evaluated on CIFAR-10 (Krizhevsky, 2009). To restrict the number of architectures in the search space, the number of nodes and edges was given an upper bound and only three operations are considered. One result of this limitation is that One-Shot NAS methods can only be applied to subspaces of NAS-Bench-101 as demonstrated in NAS-Bench-1Shot1 (Zela et al., 2020b).

NAS-Bench-201 (Dong & Yang, 2020), in contrast, uses a search space with a fixed number of nodes and edges, hence allowing for a straight-forward application of one-shot NAS methods. However, this limits the total number of unique architectures to as few as 6466. NAS-Bench-201 includes evaluations of all these architectures on three different datasets, namely CIFAR-10, CIFAR-100 (Krizhevsky, 2009) and Downsampled Imagenet 16×16 (Chrabaszcz et al., 2017), allowing for transfer learning experiments.

More recently, NATS-Bench (Dong et al., 2021) was introduced as an extension to NAS-Bench-201. It introduces a size search space for the number of channels in each cell of a cell-based macro architecture. Evaluating each configuration in their size search space for one cell, NAT-Bench provides an additional ~32k unique architectures. On the other hand, HW-NAS-Bench (Li et al., 2021b), extends the search space of NAS-Bench-201 by measuring different hardware metrics, such as latency and energy consumption on six different devices.

NAS-Bench-NLP (Klyuchnikov et al., 2020) was recently proposed as a tabular benchmark for NAS in the Natural Language Processing domain. The search space resembles NAS-Bench-101 as it limits the number of edges and nodes to constrain the search space size resulting in 14k evaluated architectures. NAS-Bench-ASR (Mehrotra et al., 2021) is the first benchmark that enables running NAS algorithms on the automatic scpeech recognition task. The benchmark is still tabular though, which restricts the search space size to around 8k unique architectures.

## A.2  NEURAL NETWORK PERFORMANCE PREDICTION

In the past, several works have attempted to predict the performance of neural networks by extrapolating learning curves (Domhan et al., 2015; Klein et al., 2017; Baker et al., 2017). A more recent line of work in performance prediction focuses more on feature encoding of neural architectures. Peephole (Deng et al., 2017) and TAPAS (Istrate et al., 2019) both use an LSTM to aggregate information about the operations in chain-structured architectures. On the other hand, BANANAS (White et al., 2021a) introduces a path-based encoding of cells that automatically resolves the computational equivalence of architectures.

Graph Neural Networks (GNNs) (Gori et al., 2005; Kipf & Welling, 2017; Zhou et al., 2018; Wu et al., 2019b; Dudziak et al., 2020), with their capability of learning representations of graph-structured data appear to be a natural choice to learning embeddings of NN architectures. Shi et al. (2019) and Wen et al. (2019) trained a Graph Convolutional Network (GCN) on a subset of NAS-Bench-101 (Ying et al., 2019) showing its effectiveness in predicting the performance of unseen architectures. Moreover, Lukasik et al. (2021) propose smooth variational graph embedding (SVGe) via a variational graph autoencoder which utilizes a two-sided GNN encoder-decoder model in the space of architectures, which generates valid graphs in the learned latent space and also allows for extrapolation.

Several recent works further adapt the GNN message passing to embed architecture bias via extra weights to simulate the operations such as in GATES (Ning et al., 2020) or integrate additional information on the operations (e.g., flop count) (Xu et al., 2019b). Tang et al. (2020) chose to operate GNNs on relation graphs based on architecture embeddings in a metric learning setting, allowing to pose NAS performance prediction as a semi-supervised setting.

Finally, White et al. (2021b) evaluate more than 30 different performance predictors on existing NAS benchmarks, ranging from model-based ones to zero-cost proxies and learning curve predictors.

## A.3 COMPARISON BETWEEN NAS AND HPO SURROGATE BENCHMARKS

Finally, we highlight the similarities and differences of surrogate benchmarks for NAS and hyper-parameter optimization (HPO) (Eggensperger et al., 2015; Klein et al., 2019). Ultimately, the NAS problem can indeed be formulated as an HPO problem and some algorithms designed for HPO can indeed be evaluated on NAS benchmarks; however, this does require that they can handle the corresponding high-dimensional categorical hyperparameter space typical of NAS benchmarks (e.g., 34 hyperparameters for the DARTS space; 22 hyperparameters for the FBNet space). This difference in dimensionality makes a qualitative difference: in HPO, a benchmark with few hyperparameters makes a lot of sense, as that is actually the most frequent use case of HPO. But this is not the case for NAS, which typically consists of much higher-dimensional spaces. In this work, we gave several existence proofs that it is indeed possible to build good-enough surrogate benchmarks for high-dimensional NAS spaces. We also showed for the first time that surrogate benchmarks can have lower error than tabular benchmarks and demonstrated an exemplary use of surrogate NAS benchmarks to drive research hypotheses in a case study; neither of these has been done for surrogate HPO benchmarks. Furthermore, we also study the possibility of using pure random search to generate the architectures to base our surrogate on; this turned out to work very well for NAS surrogate benchmarks but would work very poorly for typical higher-dimensional HPO surrogate benchmarks, because many hyperparameters, when set poorly, completely break performance, whereas poor architectural choices still yielded relevant results that could inform the model building. Finally, NAS benchmarks are also far more expensive than typical HPO benchmarks. This calls even more for the necessity of a careful methodology on how to construct surrogate benchmarks for NAS.

## B TRAINING DETAILS FOR THE GIN IN THE MOTIVATION

We set the GIN to have a hidden dimension of 64 with 4 hidden layers resulting in around ∼40k parameters. We trained for 30 epochs with a batch size of 128. We chose the MSE loss function and add a logarithmic transformation to emphasize the data fit on well-performing architectures.

| Model | Mean Squared Error (MSE) | | | Kendall tau | | |
|---|---|---|---|---|---|---|
| | 1, [2, 3] | 2, [1, 3] | 3, [1, 2] | 1, [2, 3] | 2, [1, 3] | 3, [1, 2] |
| Tab. | 5.44$e$-5 | 5.43$e$-5 | 5.34$e$-5 | 0.83 | 0.83 | 0.83 |
| Surr. | **3.02e-5** | **3.07e-5** | **3.02e-5** | **0.87** | **0.87** | **0.87** |

Table 4: MSE and Kendall tau correlation between performance predicted by a tab./surr. benchmark fitted with one seed each, and the true performance of evaluations with the two other seeds (see Section 2). Test seeds in brackets.

## C THE SURR-NAS-BENCH-DARTS DATASET

### C.1 SEARCH SPACE

We use the same architecture search space as in DARTS (Liu et al., 2019). Specifically, the normal and reduction cell each consist of a DAG with 2 input nodes (receiving the output feature maps from the previous and previous-previous cell), 4 intermediate nodes (each adding element-wise feature maps from two previous nodes in the cell) and 1 output node (concatenating the outputs of all intermediate nodes). Input and intermediate nodes are connected by directed edges representing one of the following operations: Sep. conv $3 \times 3$, Sep. conv $5 \times 5$, Dil. conv $3 \times 3$, Dil. conv $5 \times 5$, Max pooling $3 \times 3$, Avg. pooling $3 \times 3$, Skip connection.

### C.2 DATA COLLECTION

To achieve good global coverage, we use random search to evaluate ∼23k architectures. We note that space-filling designs such as quasi-random sequences, e.g. Sobol sequences (Sobol', 1967), or Latin Hypercubes (McKay et al., 2000) and Adaptive Submodularity (Golovin & Krause, 2011) may also provide good initial coverage.

Random search is supplemented by data which we collect from running a variety of optimizers, representing Bayesian Optimization (BO), evolutionary algorithms and One-Shot Optimizers. We used Tree-of-Parzen-Estimators (TPE) (Bergstra et al., 2011) as implemented by Falkner et al. (2018) as a baseline BO method. Since several recent works have proposed to apply BO over combinatorial spaces (Oh et al., 2019; Baptista & Poloczek, 2018) we also used COMBO (Oh et al., 2019). We included BANANAS (White et al., 2021a) as our third BO method, which uses a neural network with a path-based encoding as a surrogate model and hence scales better with the number of function evaluations. As two representatives of evolutionary approaches to NAS, we chose Regularized Evolution (RE) (Real et al., 2019) as it is still one of the state-of-the art methods in discrete NAS and Differential Evolution (Price et al., 2006) as implemented by Awad et al. (2020). Accounting

| | NAS methods | # eval |
|---|---|---|
| | RS (Bergstra & Bengio, 2012) | 23746 |
| Evolution | DE (Awad et al., 2020) | 7275 |
| | RE (Real et al., 2019) | 4639 |
| BO | TPE (Bergstra et al., 2011) | 6741 |
| | BANANAS (White et al., 2021a) | 2243 |
| | COMBO (Oh et al., 2019) | 745 |
| One-Shot | DARTS (Liu et al., 2019) | 2053 |
| | PC-DARTS (Xu et al., 2020) | 1588 |
| | DrNAS (Chen et al., 2021) | 947 |
| | GDAS (Dong & Yang, 2019) | 234 |

Table 5: NAS methods used to cover the search space and the number of architectures (not necessarily unique) gathered by them during search.

for the surge in interest in One-Shot NAS, our collected data collection also entails evaluation of architectures from search trajectories of DARTS (Liu et al., 2019), GDAS (Dong & Yang, 2019), DrNAS (Chen et al., 2021) and PC-DARTS (Xu et al., 2020). For details on the architecture training details, we refer to Section C.6.

For each architecture $a \in \mathcal{A}$, the dataset contains the following metrics: train/validation/test accuracy, training time and number of model parameters.

## C.3 DETAILS ON EACH OPTIMIZER

In this section we provide the hyperparameters used for the evaluations of NAS optimizers for the collection of our dataset. Many of the optimizers require a specialized representation to function on an architecture space because most of them are general HPO optimizers. As recently shown by White et al. (2020b), this representation can be critical for the performance of a NAS optimizer. Whenever the representation used by the Optimizer did not act directly on the graph representation, such as in RE, we detail how we represented the architecture for the optimizer. All optimizers were set to optimize the validation error.

**BANANAS.** We initialized BANANAS with 100 random architectures and modified the optimization of the surrogate model neural network, by adding early stopping based on a 90%/10% train/validation split and lowering the number of ensemble models to be trained from 5 to 3. These changes to bananas avoided a computational bottleneck in the training of the neural network.

**COMBO.** COMBO only attempts to maximize the acquisition function after the entire initial design (100 architectures) has completed. For workers which are done earlier, we sample a random architecture, hence increasing the initial design by the number of workers (30) we used for running the experiments. The search space considered in our work is larger than all search spaces evaluated in COMBO (Oh et al., 2019) and we regard not simply binary architectural choices, as we have to make choices about pairs of edges. Hence, we increased the number of initial samples for ascent acquisition function optimization from 20 to 30. Unfortunately, the optimization of the GP already became the bottleneck of the BO after around 600 function evaluations, leading to many workers waiting for new jobs to be assigned.

*Representation:* In contrast to the COMBO's original experimental setting, the DARTS search requires choices based on pairs of parents of intermediate nodes where the number of choices increase with the index of the intermediate nodes. The COMBO representation therefore consists of the graph cartesian product of the combinatorial choice graphs, increasing in size with each intermediate node. In addition, there exist 8 choices over the number of parameters for the operation in a cell.

**Differential Evolution.** DE was started with a generation size of 100. As we used a parallelized implementation, the workers would have to wait for one generation plus its mutations to be completed for selection to start. We decided to keep the workers busy by training randomly sampled architectures in this case, as random architectures provide us good coverage of the space. However, different

methods using asynchronous DE selection would also be possible. Note, that the DE implementation by Awad et al. (2020), performs boundary checks and resamples components of any individual that exceeds 1.0. We use the rand1 mutation operation which generally favors exploration over exploitation.

*Representation:* DE uses a vector representation for each individual in the population. Categorical choices are scaled to lie within the unit interval [0, 1] and are rounded to the nearest category when converting back to the discrete representation in the implementation by Awad et al. (2020). Similarly to COMBO, we represent the increasing number of parent pair choices for the intermediate nodes by interpreting the respective entries to have an increasing number of sub-intervals in [0, 1].

**DARTS, GDAS, PC-DARTS and DrNAS.**    We collected the architectures found by all of the above one-shot optimizers with their default search hyperparameters. We performed multiple searches for each one-shot optimizer.

**RE.**    To allow for a good initial coverage before mutations start, we decided to randomly sample 3000 architectures as initial population. RE then proceeds with a sample size of 100 to extract well performing architectures from the population and mutates them. During mutations RE first decides whether to mutate the normal or reduction cell and then proceeds to perform either a parent change, an operation change or no mutation.

**TPE.**    For TPE we use the default settings as also used by BOHB. We use the Kernel-Density-Estimator surrogate model and build two models where the good configs are chosen as the top 15%. The acquisition function's expected improvement is optimized by sampling 64 points.

**HEBO.**    Heteroscedastic and Evolutionary Bayesian Optimisation solver (HEBO) is the winning optimizer of the NeurIPS 2020 Black-Box Optimisation challenge [6]. HEBO was designed after careful observations of improvements that BO components yield on a variety of benchmarks. In particular, HEBO uses an enhanced surrogate model which can handle non-stationarity and heteroscedasticity during the marginal likelihood maximization. In our experiments we use a random forest surrogate model instead of the default Gaussian Process one, while the other components of HEBO remain unchanged with their respective default values.

### C.4    OPTIMIZER PERFORMANCE

The trajectories from the different NAS optimizers yield quite different performance distributions. This can be seen in Figure 7 which shows the ECDF of the validation errors of the architectures evaluated by each optimizer. As the computational budgets allocated to each optimizer vary widely, this data does not allow for a fair comparison between the optimizers. However, it is worth mentioning that the evaluations of BANANAS feature the best distribution of architecture performances, followed by PC-DARTS, DrNAS, DE, GDAS, and RE. TPE only evaluated marginally better architectures than RS, while COMBO and DARTS evaluated the worst architectures.

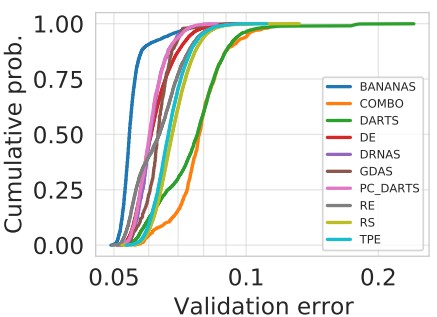

Figure 7: Empirical Cumulative Density Function (ECDF) plot comparing all optimizers. Optimizers which cover good regions of the search space feature higher values in the low validation error region.

We also perform a t-SNE analysis on the data collected by the different optimizers in Figure 13. We find that RE discovers well-performing architectures which form clusters distinct from the architectures found via RS. We observe that COMBO searched previously unexplored areas of the search space. BANANAS, which found some of the best architectures, explores clusters outside the main cluster. However, it heavily exploits regions at the cost of exploration. We argue that this is a result of the optimization of the acquisition function via random mutations based on the previously found iterates, rather than on new random architectures. DE is the only optimizer which finds well performing architectures in the center of the embedding space.

---

[6] https://bbochallenge.com/leaderboard/

## C.5 Influence of cell topology and operations

In this section, we investigate the influence of the cell topology and the operations on the performance of the architectures in our setting. The discovered properties of the search space inform our choice of metrics for the evaluation of different surrogate models.

First, we study how the validation error depends on the depth of architectures. Figure 9 visualizes the performance distribution of normal and reduction cells of different depth[7] by approximating empirical distributions with a kernel density estimation used in violin plots (Hwang et al., 1994). We observe that the performance distributions are similar for the normal and reduction cells with the same cell depth. Although cells of all depths can reach high performances, shallower cells seem slightly favored. Note that these observations are subject to changes in the hyperparameter setting, e.g. training for more epochs may render deeper cells more competitive. The best-found architecture features a normal and reduction cell of depth 4. Color-coding the cell depth in our t-SNE

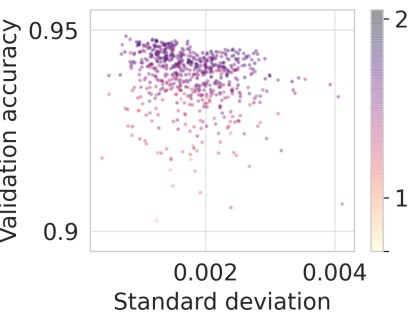

Figure 8: Standard deviation of the val. accuracy for multiple architecture evaluations.

projection also confirms that the t-SNE analysis captures the cell depth well as a structural property (c.f. Figure 11). It also reinforces that the search space is well-covered.

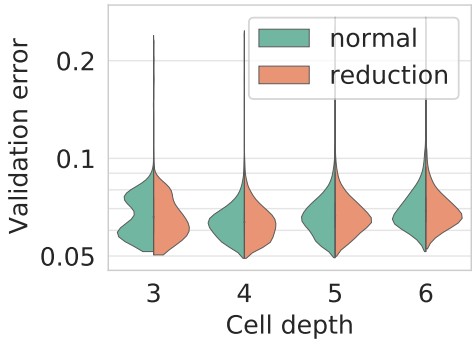

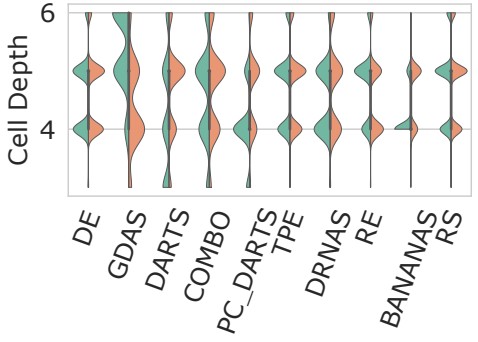

Figure 9: Distribution of the validation error for different cell depth.

Figure 10: Comparison between the normal and reduction cell depth for the architectures found by each optimizer.

We also show the distribution of normal and reduction cell depths of each optimizer in Figure 10 to get a sense for the diversity between the discovered architectures. We observe that DARTS and BANANAS generally find architectures with a shallow reduction cell and a deeper normal cell, while the reverse is true for RE. DE, TPE, COMBO and RS appear to find normal and reduction cells with similar cell depth.

Aside from the cell topology, we can also use our dataset to study the influence of operations to the architecture performance. The DARTS search space contains operation choices without parameters such as Skip-Connection, Max Pooling $3 \times 3$ and Avg Pooling $3 \times 3$. We visualize the influence of these parameter-free operations on the validation error in the normal and reduction cell in Figure 21a, respectively Figure 14. While pooling operations in the normal cell seem to have a negative impact on performance, a small number of skip connections improves the overall performance. This is somewhat expected, since the normal cell is dimension preserving and skip connections help training by improving gradient flow like in ResNets (He et al., 2016). In the reduction cell, the number of parameter-free operations has less effect as shown in Figure 14. In contrast to the normal cell

---

[7]We follow the definition of cell depth used by Shu et al. (2020), i.e. the length of the longest simple path through the cell.

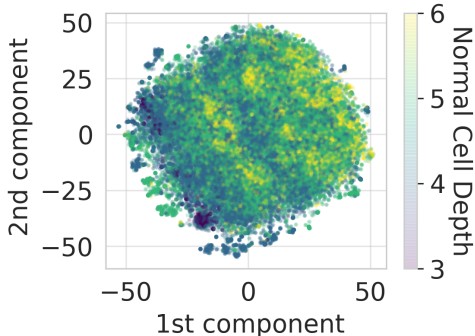
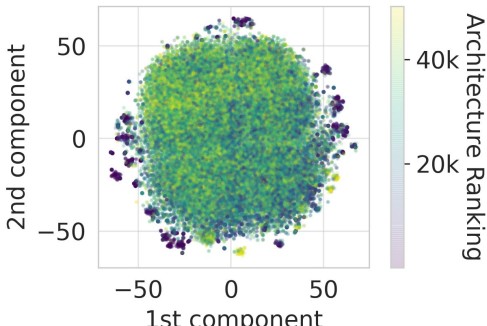

Figure 11: t-SNE projection colored by the depth of the normal cell.

Figure 12: t-SNE visualization of the sampled architectures ranked by validation accuracy.

where 2-3 skip-connections lead to generally better performance, the reduction cell shows no similar trend. For both cells, however, featuring many parameter-free operations significantly deteriorates performance. We therefore expect that a good surrogate also models this case as a poorly performing region.

## C.6 TRAINING DETAILS

Each architecture was evaluated on CIFAR-10 (Krizhevsky, 2009) using the standard 40k, 10k, 10k split for train, validation and test set. The networks were trained using SGD with momentum 0.9, initial learning rate of 0.025 and a cosine annealing schedule (Loshchilov & Hutter, 2017), annealing towards $10^{-8}$.

We apply a variety of common data augmentation techniques which differs from previous NAS benchmarks where the training accuracy of many evaluated architectures reached 100% (Ying et al., 2019; Dong & Yang, 2020) indicating overfitting on the training set. We used CutOut (DeVries & Taylor, 2017) with cutout length 16 and MixUp (Zhang et al., 2018) with alpha 0.2. For regularization, we used an auxiliary tower (Szegedy et al., 2015) with a weight of 0.4 and DropPath (Larsson et al., 2017) with drop probability of 0.2. We trained each architecture for 100 epochs with a batch size of 96, using 32 initial channels and 8 cell layers. We chose these values to be close to the proxy model used by DARTS while also achieving good performance.

## C.7 NOISE IN ARCHITECTURE EVALUATIONS

As discussed in Section 2, the noise in architecture evaluations can be large enough for surrogate models to yield more realistic estimates of architecture performance than a tabular benchmark based on a single evaluation per architecture. To study the magnitude of this noise on Surr-NAS-Bench-DARTS, we evaluated 500 architectures randomly sampled from our Differential Evolution (DE) (Awad et al., 2020) run with 5 different seeds each. We chose DE because it both explored and exploited well; see Figure 13. We find a mean standard deviation of 1.6e−3 for the final validation accuracy, which is slightly less than the noise observed in NAS-Bench-101 (Ying et al., 2019); one possible reason for this could be a more robust training pipeline. Figure 8 shows that, while the noise

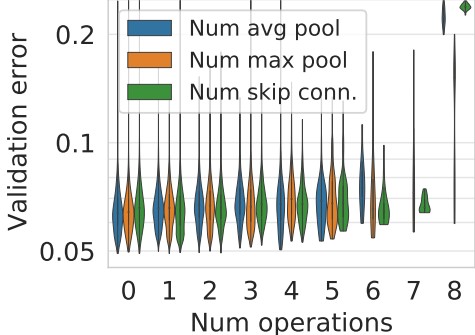

Figure 14: Distribution of validation error in dependence of the number of parameter-free operations in the reduction cell. Violin plots are cut off at the respective observed minimum and maximum value.

tends to be lower for the best architectures, a correct ranking based on a single evaluation is still difficult. Finally, we compare the MAE when estimating the architecture performance from only one

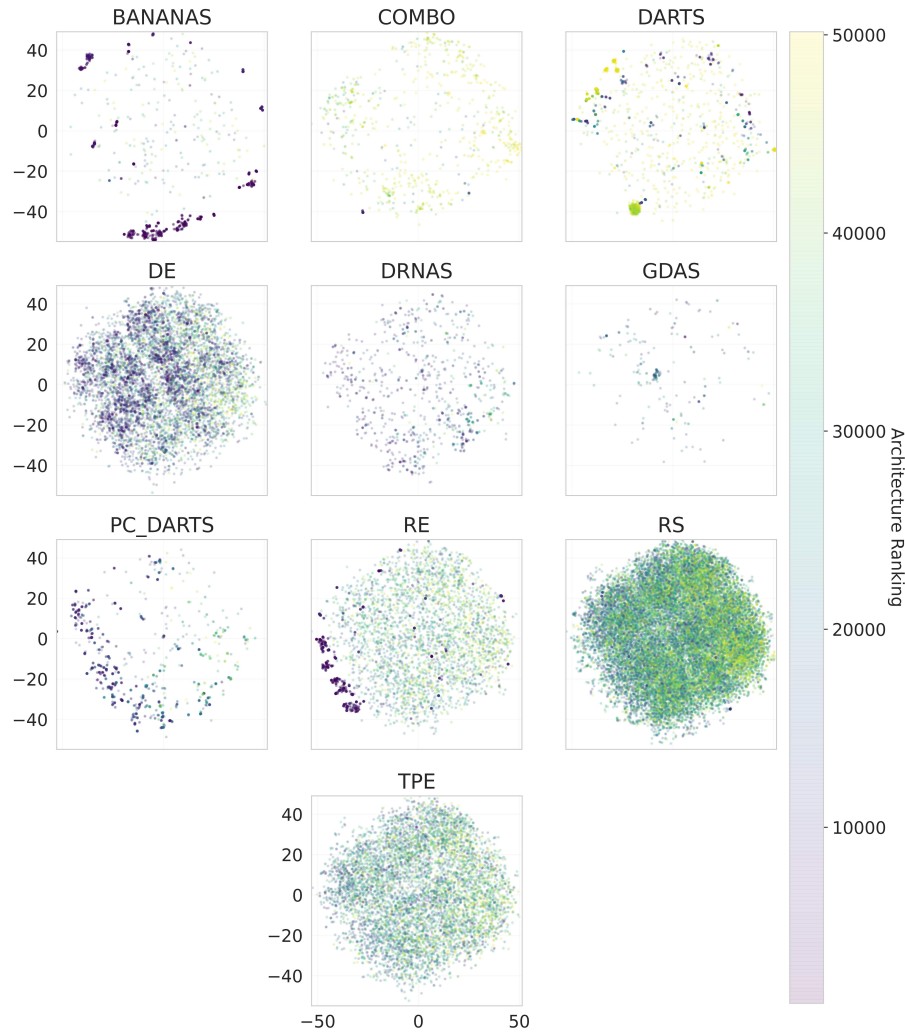

Figure 13: Visualization of the exploration of different parts of the architectural t-SNE embedding space for all optimizers used for data collection. The architecture ranking by validation accuracy (lower is better) is global over the entire data collection of all optimizers.

sample to the results from Table 1. Here, we also find a slightly lower MAE of $1.38e-3$ than for NAS-Bench-101.

## D   THE SURR-NAS-BENCH-FBNET DATASET

### D.1   SEARCH SPACE

The FBNet (Wu et al., 2019a) search space consists of a sequence of 26 stacked layers, out of which 22 need to be searched. The first and last three layers have fixed operations, while the rest can have one out of a total of 9 candidate blocks. These include a skip connection "block" and the rest of the block choices are inspired by the MobileNetV2 (Sandler et al., 2018) and ShiftNet (Wu et al., 2018). Their strructure contains a point-wise $1 \times 1$ convolution, a $K \times$ K depthwise convolution where K denotes the kernel size, and another 1x1 convolution. ReLU is applied only after the first 1x1 convolution and the depthwise convolution, but not after the last 1x1 convolution. A skip connection adding the input to the output is used if the output dimension remains the same as the input one. Overall, the

block choices are: $\{k3\_e1, k3\_e1_g2, k3\_e3, k3\_e6, k5\_e1, k5\_e1_g2, k5\_e3, k5\_e6, skip\}$, where $k$ denotes the kernel size, $e$ the expansion ratio and $g$ if group convolution is used.

### D.2 TRAINING DETAILS

Each architecture was evaluated on CIFAR-100 (Krizhevsky, 2009) using the standard 40k, 10k, 10k split for train, validation and test set. The networks were trained using SGD with momentum 0.9, batch size 256, initial learning rate of 0.025, $L_2$ regularization with 0.0005 coefficient and a cosine annealing (Loshchilov & Hutter, 2017) schedule towards 0. We furthermore apply sharpness-aware minimization (Foret et al., 2021) for better generalization. We also apply label smoothing with a smoothing factor of 0.1.

## E SURROGATE MODEL ANALYSIS

### E.1 PREPROCESSING OF THE GRAPH TOPOLOGY

**DGN preprocessing** All DGN were implemented using PyTorch Geometric (Fey & Lenssen, 2019) which supports the aggregation of edge attributes. Hence, we can naturally represent the DARTS architecture cells, by assigning the embedded operations to the edges. The nodes are labeled as input, intermediate and output nodes. We represent the DARTS graph as shown in Figure 15, by connecting the output node of each cell type with the inputs of the other cell, allowing information from both cells to be aggregated per node during message passing. Note the self-loop on the output node of the normal cell, which we found necessary to get the best performance.

**Preprocessing for other surrogate models** Since we make use of the framework implemented by BOHB (Falkner et al., 2018) to easily parallelize the architecture search algorithms across many compute nodes, we also represent our search space using ConfigSpace [8] (Lindauer et al., 2019). More precisely, we encode each pair of incoming edges for a cell as one choice of a categorical parameter. For instance, for node 4 in the normal cell, we add a parameter `inputs_node_normal_4` with the choices of edge pairs `0_1,0_2,0_3,1_2,1_3,2_3`. The edge operations are then implemented as categorical parameters for each edge and are only active if the corresponding edge was chosen. For instance, in the example above, if the incoming edge 0 is sampled, the parameter associated with the edge from node 0 to node 4 becomes activate and one operation is sampled. We provide the configuration space with our code. For all non-DGN based surrogate models, we use the vector representation of a configuration given by ConfigSpace as input to the model. This vector representation contains one value between 0 and 1 for each parameter in the configuration space.

### E.2 DETAILS ON THE GIN

The GIN implementation on the Open Graph Benchmark (OGB) (Hu et al., 2020) uses virtual nodes (additional nodes which are connected to all nodes in the graph) to boost performance as well as generalization and consistently achieves good performance on their public leaderboards. Other GNNs from Errica et al. (2020), such as DGCNN and DiffPool, performed worse in our initial experiments and are therefore not considered.

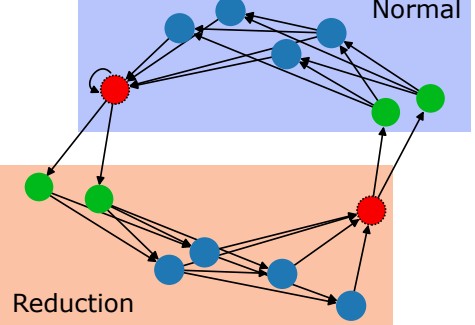

Figure 15: Architecture with inputs in green, intermediate nodes in blue and outputs in red.

Following recent work in Predictor-based NAS (Ning et al., 2020; Xu et al., 2019b), we use a per batch ranking loss because the ranking of an architecture is equally important to an accurate prediction of the validation accuracy in a NAS setting. We use the ranking loss formulation by GATES (Ning et al., 2020) which is a hinge pair-wise ranking loss with margin m=0.1.

---

[8] https://github.com/automl/ConfigSpace

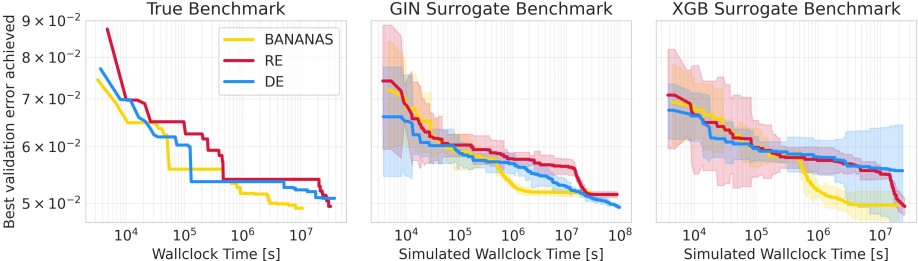

Figure 16: Anytime performance of blackbox optimizers, comparing performance achieved on the real benchmark and on surrogate benchmarks built with GIN and XGB in an LOOO fashion.

### E.3    DETAILS ON HPO

All detailed table for the hyperparameter ranges for the HPO and the best values found by BOHB are listed in Table 6. We used a fixed budget of 128 epochs for the GNN, a minimum budget of 4 epochs and maximum budget of 128 epochs for the MLP and a fixed budget for all remaining model.

### E.4    HPO FOR RUNTIME PREDICTION MODEL

Our runtime prediction model is an LGB model trained on the runtimes of architecture evaluations of DE. This is because we partially evaluated the architectures utilizing different CPUs. Hence, we only choose to train on the evaluations carried out by the same optimizer on the same hardware to keep a consistent estimate of the runtime. DE is a good choice in this case because it both explored and exploited the architecture space well. The HPO space used for the LGB runtime model is the same used for the LGB surrogate model.

### E.5    LEAVE ONE-OPTIMIZER-OUT ANALYSIS

The results in Table 7 for `SNB-DARTS` show that the rank correlation between the predicted and observed validation accuracy remains high even when a well-performing optimizer such as RE is left out. A detailed scatter plot of the predicted performance against the true performance for each optimizer and surrogate model in an LOOO analysis is provided in Figure 18 and Figure 19. Predicting BANANAS in the LOOO fashion yields a lower rank correlation, because it predominantly selects well-performing architectures, which are harder to rank; however, the high $R^2$ shows that the fit is still good. Conversely, leaving out DARTS causes a low $R^2$ but still high rank correlation; this is due to architectures with many skip connections in the DARTS data that are overpredicted.

The surrogates extrapolate well enough to these regions to identify that they contain poor architectures that rank worse than those in other areas of the space (which is the most important property we would like to predict correctly); but without having seen training data of extremely poor architectures with many skip connections they cannot predict *how bad exactly* these architectures are.

This can be improved by fitting the surrogate on data that contains more such architectures, something we study further in Appendix F.1.

To avoid poor predictions in such boundary cases, in our guidelines for creating new surrogate NAS benchmarks (Appendix G) we recommend adding architectures from known areas of poor performance to the training data.

### E.6    PARAMETER-FREE OPERATIONS

Several works have found that methods based on DARTS (Liu et al., 2019) are prone to finding sub-optimal architectures that contain many, or even only, parameter-free operations (max. pooling, avg. pooling or skip connections) and perform poorly (Zela et al., 2020a). We therefore evaluated the surrogate models on such architectures by replacing a random selection of operations in a cell with one type of parameter-free operations to match a certain ratio of parameter-free operations in a cell. This analysis is carried out over the test set of the surrogate models and hence contains architectures

| Model | Hyperparameter | Range | Log-transform | Default Value |
|---|---|---|---|---|
| GIN | Hidden dim. | [16, 256] | true | 24 |
| | Num. Layers | [2, 10] | false | 8 |
| | Dropout Prob. | [0, 1] | false | 0.035 |
| | Learning rate | [1e-3, 1e-2] | true | 0.0777 |
| | Learning rate min. | const. | - | 0.0 |
| | Batch size | const. | - | 51 |
| | Undirected graph | [true, false] | - | false |
| | Pairwise ranking loss | [true, false] | - | true |
| | Self-Loops | [true, false] | - | false |
| | Loss log transform | [true, false] | - | true |
| | Node degree one-hot | const. | - | true |
| BANANAS | Num. Layers | [1, 10] | true | 17 |
| | Layer width | [16, 256] | true | 31 |
| | Dropout Prob. | const. | - | 0.0 |
| | Learning rate | [1e-3, 1e-1] | true | 0.0021 |
| | Learning rate min. | const. | - | 0.0 |
| | Batch size | [16, 128] | - | 122 |
| | Loss log transform | [true, false] | - | true |
| | Pairwise ranking loss | [true, false] | - | false |
| XGBoost | Early Stopping Rounds | const. | - | 100 |
| | Booster | const. | - | gbtree |
| | Max. depth | [1, 15] | false | 13 |
| | Min. child weight | [1, 100] | true | 39 |
| | Col. sample bylevel | [0.0, 1.0] | false | 0.6909 |
| | Col. sample bytree | [0.0, 1.0] | false | 0.2545 |
| | lambda | [0.001, 1000] | true | 31.3933 |
| | alpha | [0.001, 1000] | true | 0.2417 |
| | Learning rate | [0.001, 0.1] | true | 0.00824 |
| LGBoost | Early stop. rounds | const. | - | 100 |
| | Max. depth | [1, 25] | false | 18 |
| | Num. leaves | [10, 100] | false | 40 |
| | Max. bin | [100, 400] | false | 336 |
| | Feature Fraction | [0.1, 1.0] | false | 0.1532 |
| | Min. child weight | [0.001, 10] | true | 0.5822 |
| | Lambda L1 | [0.001, 1000] | true | 0.0115 |
| | Lambda L2 | [0.001, 1000] | true | 134.5075 |
| | Boosting type | const. | - | gbdt |
| | Learning rate | [0.001, 0.1] | true | 0.0218 |
| Random Forest | Num. estimators | [16, 128] | true | 116 |
| | Min. samples split. | [2, 20] | false | 2 |
| | Min. samples leaf | [1, 20] | false | 2 |
| | Max. features | [0.1, 1.0] | false | 0.1706 |
| | Bootstrap | [true, false] | - | false |
| $\epsilon$-SVR | C | [1.0, 20.0] | true | 3.066 |
| | coef. 0 | [-0.5, 0.5] | false | 0.1627 |
| | degree | [1, 128] | true | 1 |
| | epsilon | [0.01, 0.99] | true | 0.0251 |
| | gamma | [scale, auto] | - | auto |
| | kernel | [linear, rbf, poly, sigmoid] | - | sigmoid |
| | shrinking | [true, false] | - | false |
| | tol | [0.0001, 0.01] | - | 0.0021 |
| $\mu$-SVR | C | [1.0, 20.0] | true | 5.3131 |
| | coef. 0 | [-0.5, 0.5] | false | -0.3316 |
| | degree | [1, 128] | true | 128 |
| | gamma | [scale, auto] | - | scale |
| | kernel | [linear, rbf, poly, sigmoid] | - | rbf |
| | nu | [0.01, 1.0] | false | 0.1839 |
| | shrinking | [true, false] | - | true |
| | tol | [0.0001, 0.01] | - | 0.003 |

Table 6: Hyperparameters of the surrogate models and the default values found via HPO. The "Range" column denotes the ranges that the HPO algorithm used for sampling the hyperparameter values and the "Log-transform" column if sampling distribution was log-transformed or not.

| | Model | No RE | No DE | No COMBO | No TPE | No BANANAS | No DARTS | No PC-DARTS | No DrNAS | No GDAS |
|---|---|---|---|---|---|---|---|---|---|---|
| $R^2$ | LGB | **0.917** | **0.892** | **0.919** | **0.857** | 0.909 | -0.093 | **0.826** | 0.699 | 0.429 |
| | XGB | 0.907 | 0.888 | 0.876 | 0.842 | **0.911** | -0.151 | 0.817 | 0.631 | **0.672** |
| | GIN | 0.856 | 0.864 | 0.775 | 0.789 | 0.881 | **0.115** | 0.661 | **0.790** | 0.572 |
| sKT | LGB | **0.834** | **0.782** | **0.833** | **0.770** | 0.592 | **0.780** | **0.721** | 0.694 | 0.595 |
| | XGB | 0.831 | 0.780 | 0.817 | 0.762 | **0.596** | 0.775 | 0.710 | **0.709** | **0.638** |
| | GIN | 0.798 | 0.757 | 0.737 | 0.718 | 0.567 | 0.765 | 0.645 | 0.706 | 0.607 |

Table 7: Leave One-Optimizer-Out performance of the best surrogate models. Note that the configurations sampled by DARTS are mostly composed with skip connections (Zela et al., 2020a), but still the surrogate manages to rank them fairly good (high sKT) even though not providing a good estimate of the accuracy.

collected by all optimizers. For a more robust analysis, we repeated this experiment 4 times for each ratio of operations to replace.

**Results**  Figure 21 shows that both the GIN and the XGB model correctly predict that the accuracy drops with too many parameter-free operations, particularly for skip connections. The groundtruth of architectures with only parameter-free operations is displayed as scatter plot. Out of the two models, XGB captures the slight performance improvement of using a few skip connections better. LGB failed to capture this trend but performed very similarly to XGB for the high number of parameter-free operations.

### E.7  CELL TOPOLOGY ANALYSIS

Furthermore, we analyze how accurate changes in the cell topology (rather than in the operations) are modeled by the surrogates. We collected groundtruth data by evaluating all $\prod_{k=1}^{4} \frac{(k+1)k}{2} = 180$ different cell topologies (not accounting for isomorphisms) with fixed sets of operations. We assigned the same architecture to the normal and reduction cell, to focus on the effect of the cell topology. We sampled 10 operation sets uniformly at random, leading to 1800 architectures as groundtruth for this analysis.

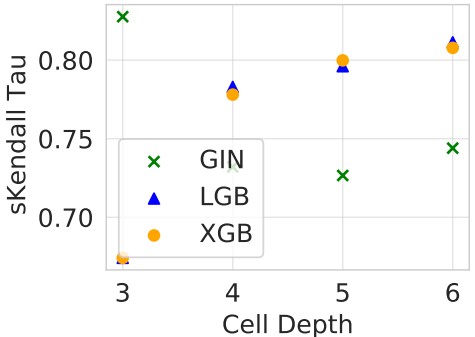

Figure 17: Comparison between GIN, XGB and LGB in the cell topology analysis.

We evaluated all architectures and group the results based on the cell depth. For each of the possible cell depths, we then computed the sparse Kendall $\tau$ rank correlation between the predicted and true validation accuracy.

**Results**  Results of the cell topology analysis are shown in Figure 17. We observe that LGB slightly outperforms XGB, both of which perform better on deeper cells. The GIN performs best for the shallowest cells.

## F  BENCHMARK ANALYSIS

### F.1  ONE-SHOT TRAJECTORIES

Surrogate NAS benchmarks, like Surr-NAS-Bench-DARTS, can also be used to monitor the behavior of one-shot NAS optimizers throughout their search phase, by querying the surrogate model with the currently most promising discrete architecture. This can be extremely useful in many scenarios since uncorrelated proxy (performance of the one-shot model) and true objectives (performance of the fully trained discretized architecture) can lead to potential failure modes, e.g., to a case where the found architectures contain only skip connections in the normal cell (Zela et al., 2020a;b; Dong & Yang, 2020) (we study such a failure case in Appendix F.1 to ensure robustness of the surrogates in said case). We demonstrate this use case in a similar LOOO analysis as for the black-box optimizers,

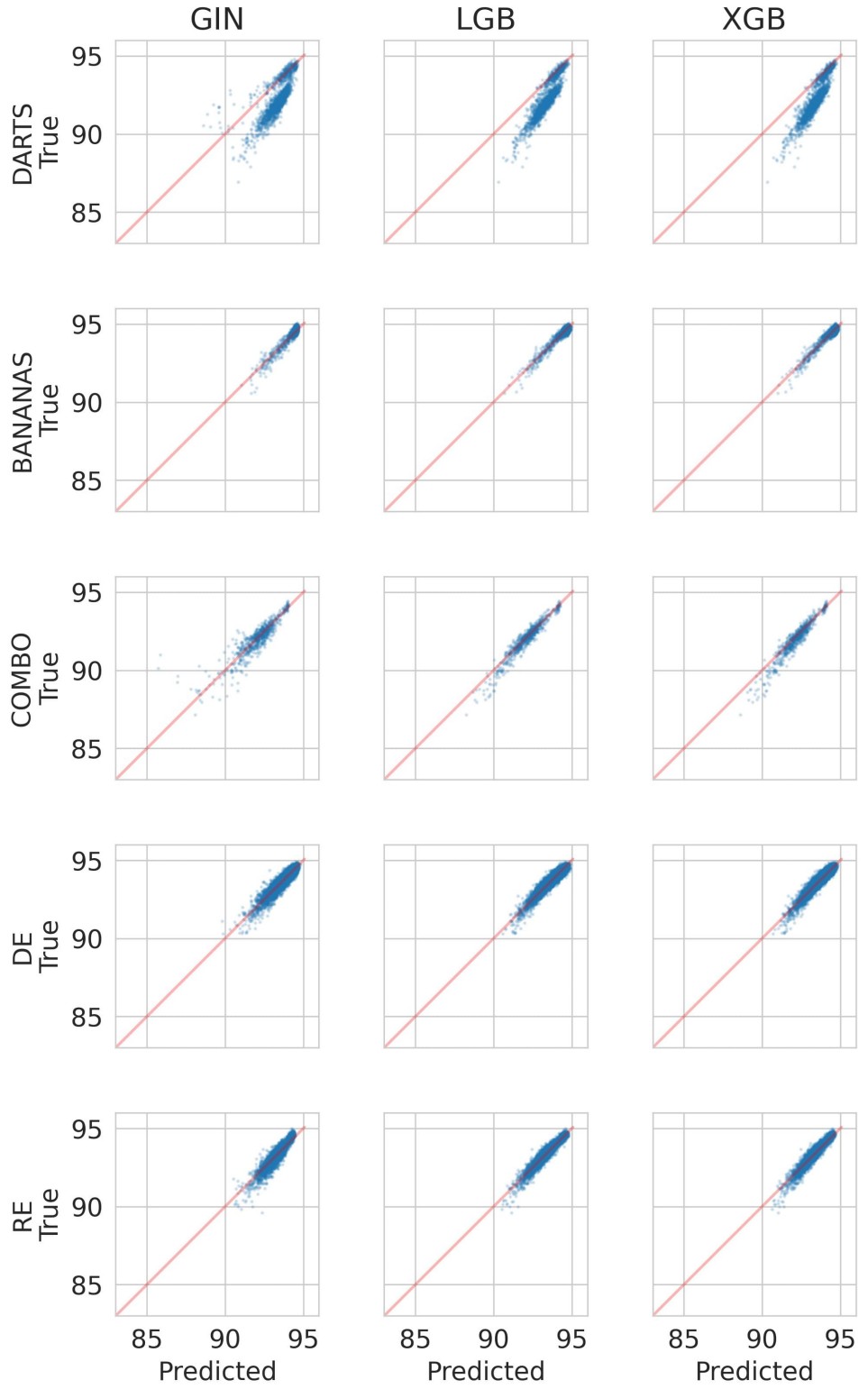

Figure 18: Scatter plots of the predicted performance against the true performance of different surrogate models on the test set in a Leave-One-Optimizer-Out setting.

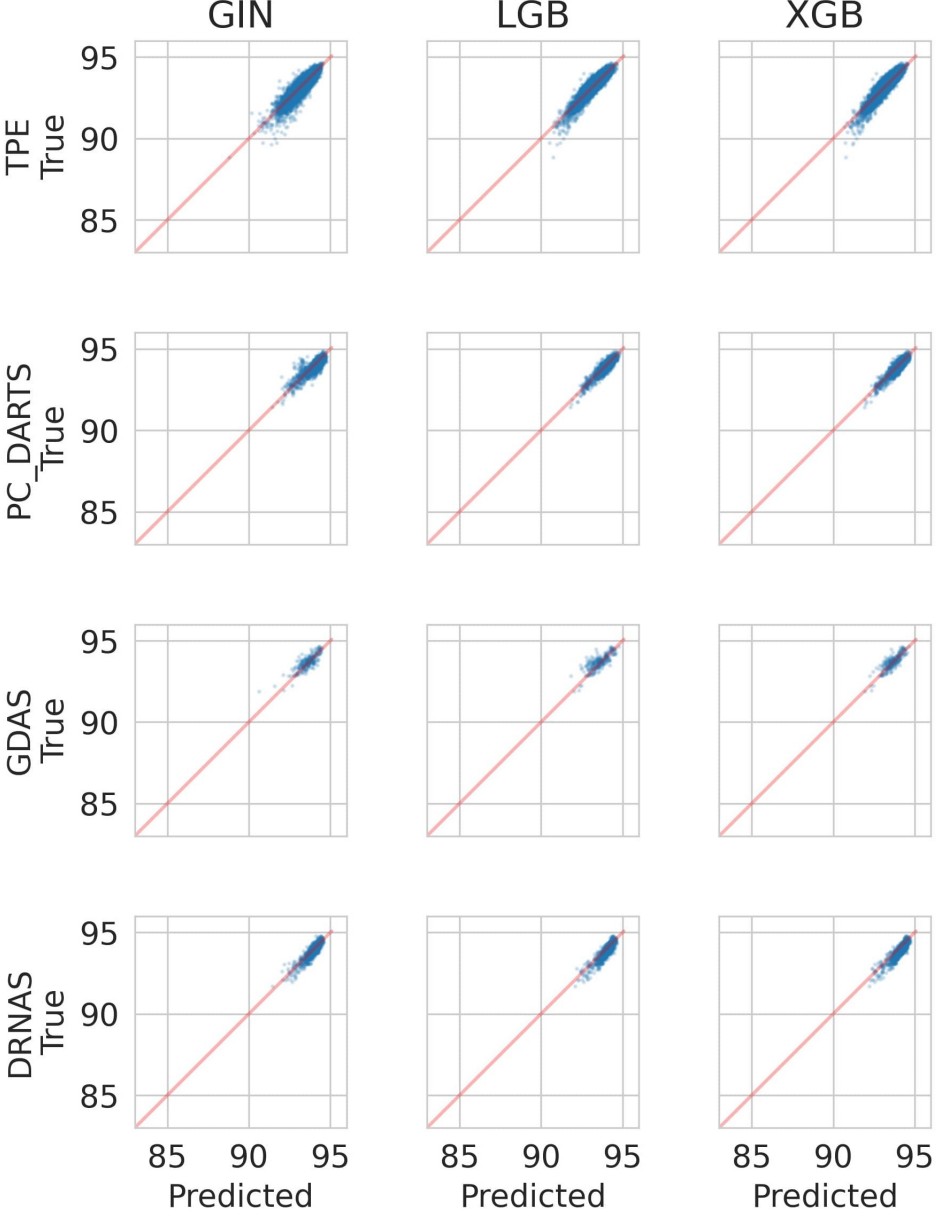

Figure 19: (continued) Scatter plots of the predicted performance against the true performance of different surrogate models on the test set in a Leave-One-Optimizer-Out setting.

using evaluations of the discrete architectures from each search epoch of multiple runs of DARTS, PC-DARTS and GDAS as ground-truth. Figure 22 shows that the surrogate trajectories closely resemble the true trajectories.

To obtain groundtruth trajectories for DARTS, PC-DARTS and GDAS, we performed 5 runs for each optimizer with 50 search epochs and evaluated the architecture obtained by discretizing the one-shot model at each search epoch. For DARTS, in addition to the default search space, we collected trajectories on the constrained search spaces from Zela et al. (2020a) to cover a failure case where DARTS diverges and finds architectures that only contain skip connections in the normal cell. To show that our benchmark is able to predict this divergent behavior, we show surrogate trajectories when training on all data, when leaving out the trajectories under consideration from the training data, and when leaving out all DARTS data in Figure 20.

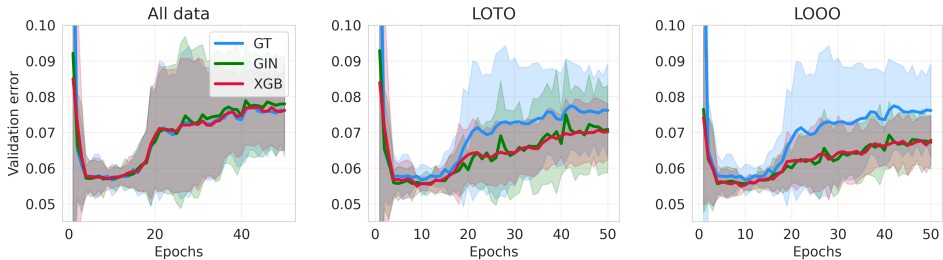

Figure 20: Ground truth (GT) and surrogate trajectories on a constrained search space where the surrogates are trained with all data, leaving out the trajectories under consideration (LOTO), and leaving out all DARTS architectures (LOOO).

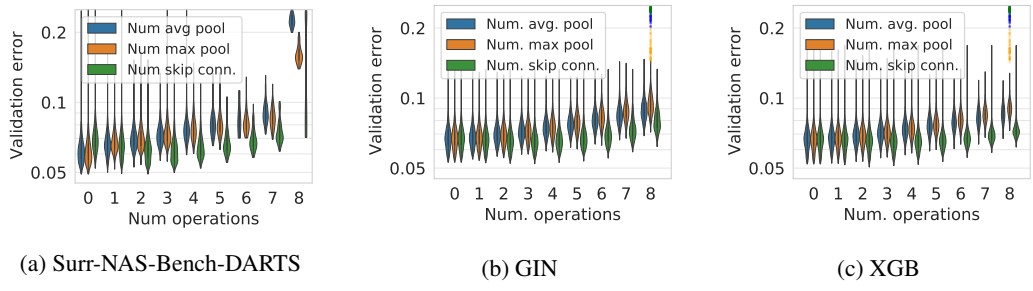

(a) Surr-NAS-Bench-DARTS    (b) GIN    (c) XGB

Figure 21: (Left) Distribution of validation error in dependence of the number of parameter-free operations in the normal cell on the Surr-NAS-Bench-DARTS dataset. (Middle and Right) Predictions of the GIN and XGB surrogate model. The collected groundtruth data is shown as scatter plot. Violin plots are cut off at the respective observed minimum and maximum value.

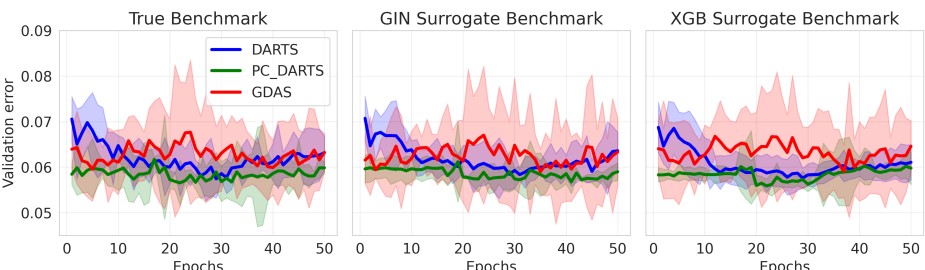

Figure 22: Anytime performance of one-shot optimizers, comparing performance achieved on the real benchmark and on surrogate benchmarks built with GIN and XGB in a LOOO fashion.

While the surrogates model the divergence in all cases, they still overpredict the architectures with only skip connections in the normal cell especially when leaving out all data from DARTS. The bad performance of these architectures is predicted more accurately when including data from other DARTS runs. This can be attributed to the fact that the surrogate models have not seen any, respectively very few data, in this region of the search space. Nevertheless, it is modeled as a bad-performing region and we expect that this could be further improved on by including additional training data accordingly, since including all data in training shows that the models are capable to of capturing this behavior.

### F.2    Ablation Study: Fitting surrogate models only on well-performing regions of the search space

To assess whether poorly-performing architectures are important for the surrogate benchmark, we fitted a GIN ensemble and an XGB ensemble model only on architectures that achieved a validation accuracy above 92%. We then tested on all architectures that achieved a validation below 92%.

Indeed, we observe that the resulting surrogate model overpredicts accuracy in regions of the space with poor performance, resulting in a low $R^2$ of -0.142 and sparse Kendall tau of 0.293 for the GIN. The results for one member of the GIN ensemble are shown in Figure 23. The XGB model achieved similar results. Next, to study whether these weaker surrogate models can still be used to benchmark NAS optimizers, we also studied optimization trajectories of NAS optimizers on surrogate benchmarks based on these surrogate models. Figure 25 shows that these surrogate models indeed suffice to accurately predict the performance achieved by Random Search and BANANAS as a function of time.

### F.3    Ablation Study: Fitting surrogate models only with random data

In this section, we would like to take the Leave-One-Optimizer-Out analysis from Section 3.1.7 one step further by leaving out *all* architectures that were collected from NAS optimizers other than random search. While the LOOO analysis removes some "bias" from the benchmark ("bias" referring to its precision in a subspace), there still is a possibility that different optimizers we used explore similar subspaces, and leaving out one of them still yields "bias" induced from a similar optimizer used for generating training data. For instance, the t-SNE analysis from Figure 13 suggests that some optimizers exploit very distinct regions (e.g., BANANAS and DE) while others exploit regions somewhat similar to others (e.g., RE and PC-DARTS). The exploration behavior, on the other hand, is quite similar across optimizers since most of them perform random sampling in the beginning. Thus, in the following, we investigate whether we can create a benchmark that has no prior information about solutions any optimizer might find.

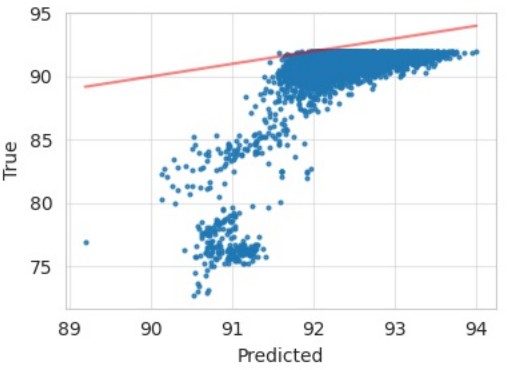

Figure 23: Scatter plot of GIN predictions on architectures that achieved below 92% validation accuracy.

To that end, we studied surrogate models based i) only on the 23746 architectures explored by random search and ii) only on 23 746 (47.3%) architectures of the original training set (sampled in a stratified manner, i.e., using 47.3% of the architectures from each of our sources of architectures).

First, we investigated the difference in the predictive performance of surrogates based on these two different types of architectures. Specifically, we fitted our GNN and XGB surrogate models on different subsets of the respective training sets and assess their predictions on unseen architectures from all optimziers as a test set. Figure 24 shows that including architectures from optimizer trajectories in the training set consistently yields significantly better generalization.

Next, we also studied the usefulness of surrogate benchmarks based on the 23 746 random architectures, compared to surrogate benchmarks based on the 23 746 architectures sampled in a stratified manner from the original set of architectures. Specifically, we used them to assess the best per-

formance achieved by various NAS optimizers as a function of time. Comparing the trajectories in Figure 3 (based on purely random architectures for training) and Figure 26 (based on 23 746 architectures sampled in a stratified manner), we find that the surrogates fitted only on random architectures work just as well as the surrogates that use architectures from NAS optimizers.

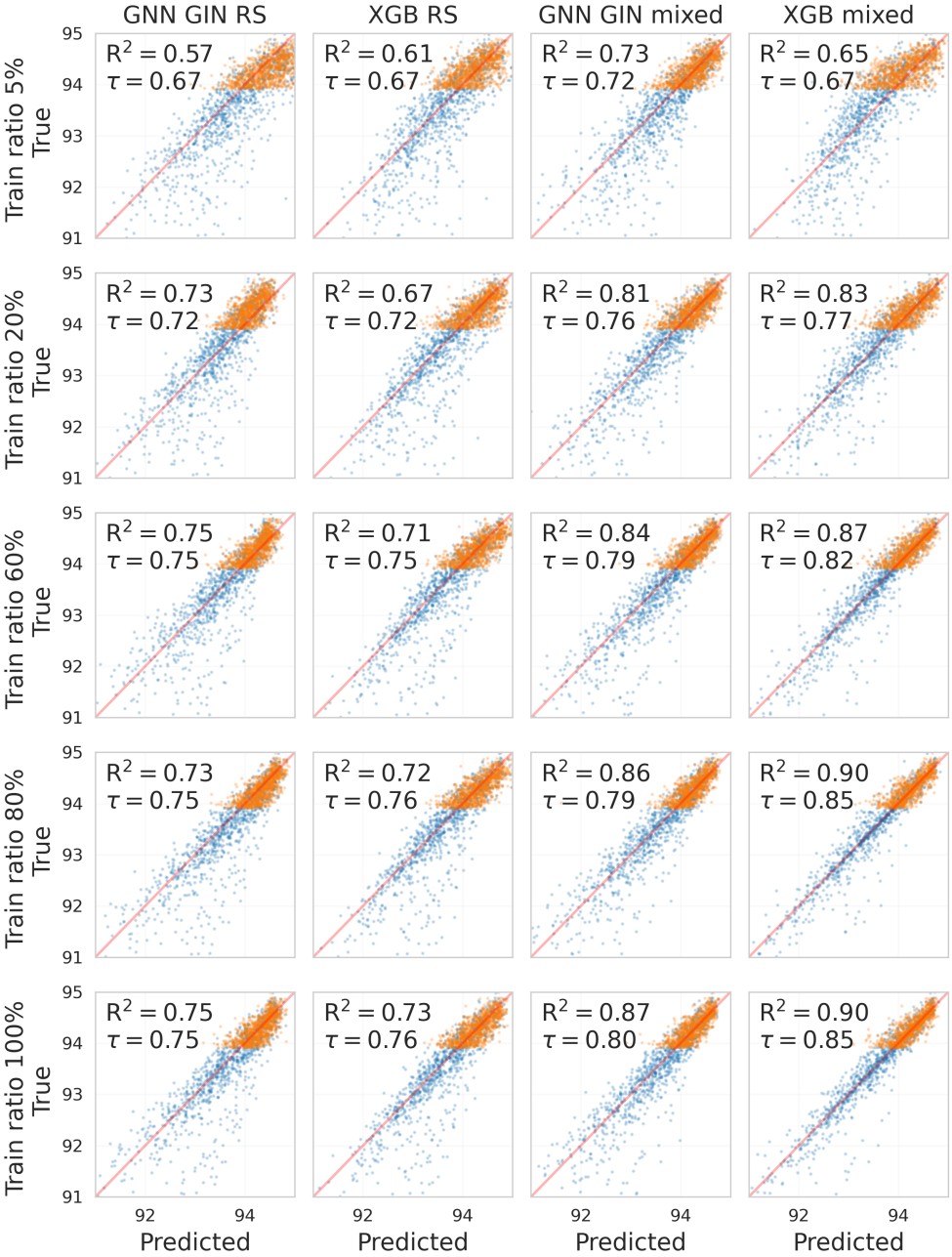

Figure 24: Scatter plots of the predicted performance against the true performance of the GNN GIN/XGB surrogate models trained with different ratios of training data. "RS" indicates that the training set only includes architectures from random search, "mixed" indicates the training set includes architectures from all optimizers. Training set sizes are identical for the two cases. The test set contains architectures from all optimizers. For better display, we show 1000 randomly sampled architectures (blue) and 1000 architectures sampled from the top 1000 architectures (orange). For each case we also show the $R^2$ and Kendall-$\tau$ coefficients on the whole test set.

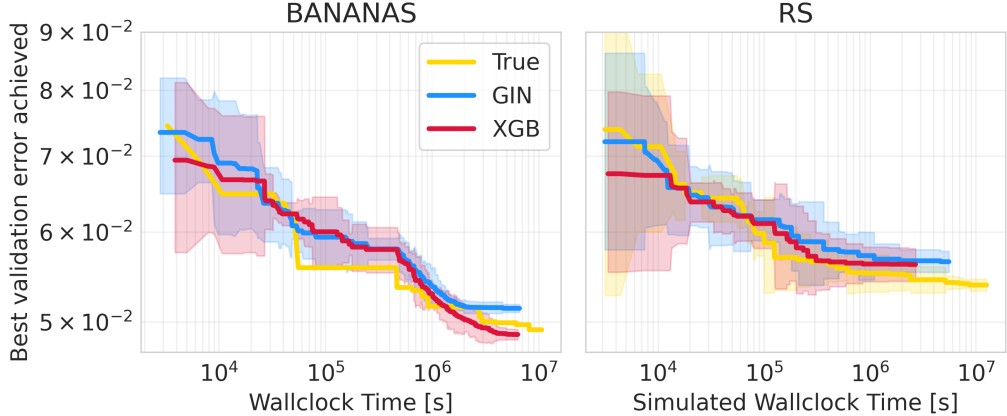

Figure 25: Comparison between the observed true trajectory of BANANAS and RS with the surrogate benchmarks only trained on well performing regions of the space

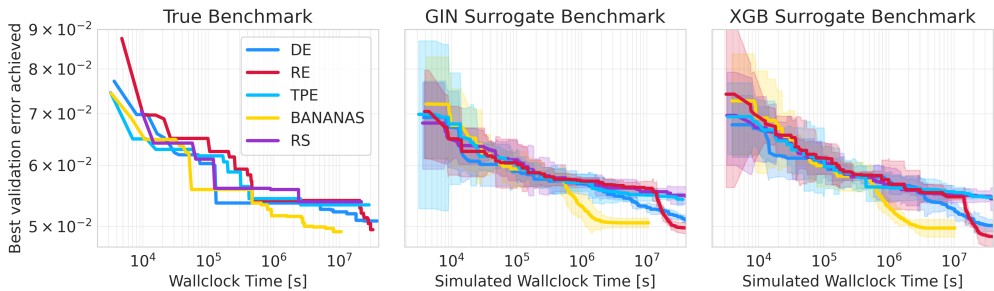

Figure 26: Anytime performance of different optimizers on the real benchmark (left) and the surrogate benchmark (GIN (middle) and XGB (right)) when training ensembles on 47.3% of the data collected from all optimizers. Trajectories on the surrogate benchmark are averaged over 5 optimizer runs.

Given this positive result for surrogates based purely on random architectures, we conclude that it is indeed possible to create surrogate NAS benchmarks that are by design free of bias towards any particular NAS optimizer (other than random search). While the inclusion of architectures generated with NAS optimizers in the training set substantially improves performance predictions of individual architectures, realistic trajectories of incumbent performance as a function of time can also be obtained with surrogate benchmarks based solely on random architectures. We note that the "unbiased" benchmark could possibly be further improved by utilizing more sophisticated space-filling sampling methods, such as the ones mentioned in Appendix C.2, or by deploying surrogate models that extrapolate well.

## F.4 EVALUATING THE SURROGATE BENCHMARK BUILT ON THE NAS-BENCH-101 DATA

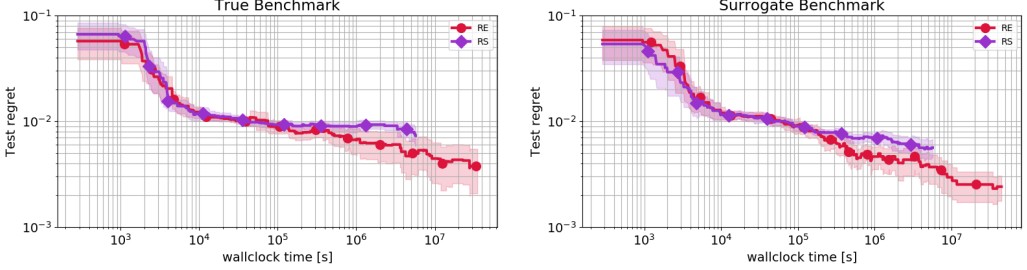

Figure 27: Anytime performance of RE and RS on the tabular NAS-Bench-101 benchmark (left) and on the surrogate benchmark version of it using the GIN model (right).

In this section we run both regularized evolution (RE) and random search (RS) on the tabular NAS-Bench-101 (Ying et al., 2019) benchmark and on the surrogate version of it, that we construct by fitting the GIN surrogate on a subset of the data available in NAS-Bench-101. Note that we had to make some changes to the GIN in order to be consistent with the graph representation in NAS-Bench-101, i.e. operations being in nodes, rather than in the edges of the graph. We also did tune the hyperparameters of the GIN on a separate validation set using BOHB (Falkner et al., 2018). In Figure 27 we plot the RE and RS incumbent trajectories, when ran on both the tabular NAS-Bench-101 benchmark (left plot) and on the GIN surrogate version of it (right plot). The y axis shows the test regret of the incumbent trajectories. As we can see, even though the performance is slightly overestimated by the surrogate benchmark in the very end of the curves, the ranking is still preserved.

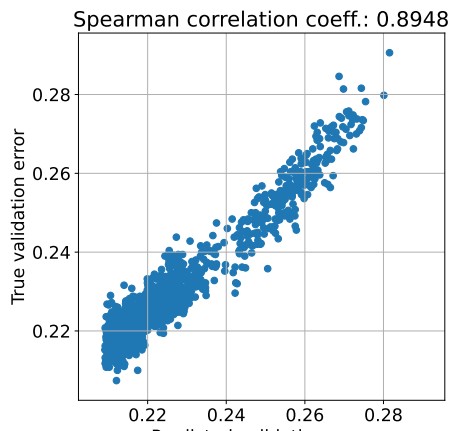

Figure 28: Scatter plot showing the predicted validation error by the XGBoost model of the configurations in the incumbent trajectory of one RE run (1000 function evaluations) on `SNB-FBNet` (x axis) vs. the true validation error values of the same configurations when retrained from scratch (y axis).

### F.5 RETRAINING THE INCUMBENT TRAJECTORIES ON THE SURROGATE BENCHMARK

In Figure 28 we plot the predicted validation error by the XGBoost model of the configurations in the RE incumbent trajectory (one of the red lines in Figure 5, right plot) on `SNB-FBNet` vs. the true validation error values of the same configurations when retrained from scratch. As one can see the ranking is still preserved (Kendall Tau correlation coefficient of 0.73 and Spearman rank correlation coefficient of 0.895) even though the performance is slightly overestimated for some configurations at the end of the trajectory.

## G   GUIDELINES FOR CREATING SURROGATE BENCHMARKS

In order to help with the design of realistic surrogate benchmarks in the future, we provide the following list of guidelines:

- Data Collection: The data collected for the NAS benchmark should provide (1) a good overall coverage, (2) explore strong regions of the space well, and (3) optimally also cover special areas in which poor generalization performance may otherwise be expected. We would like to stress that depending on the search space, a good overall coverage may already be sufficient to correctly assess the ranking of different optimizers, but as shown in Appendix F.3 additional architectures from strong regions of the space allow to increase the fidelity of the surrogate model.

  1. A good overall coverage can be obtained by random search (as in our case), but one could also imagine using better space-filling designs or adaptive methods for covering the space even better. In order to add additional varied architectures, one could also think about fitting one or more surrogate models to the data collected thus far, finding the regions of maximal predicted uncertainty, evaluate architectures there and add them to the collected data, and iterate. This would constitute an active learning approach.

  2. A convenient and efficient way to identify regions of strong architectures is to run NAS methods. In this case, the found regions should not only be based on the strong architectures one NAS method finds but rather on a set of strong and varied NAS methods (such as, in our case, one-shot methods and different types of discrete methods, such as Bayesian optimization and evolution). In order to add additional strong architectures, one could also think about fitting one or more several surrogate models to the data collected thus far, finding the predicted optima of these models, evaluate and add them to the collected data and iterate. This would constitute a special type of Bayesian optimization.

  3. Special areas in which poor generalization performance may otherwise be expected may, as in our case, e.g., include architectures with many parameterless connections, and in particular, skip connections. Other types of encountered failure modes would also be useful to cover.

- Surrogate Models: As mentioned in the guidelines for using a surrogate benchmark (see Section 5), benchmarking an algorithm that internally uses the same model type as the surrogate model should be avoided. Therefore, to provide a benchmark for a diverse set of algorithms, we recommend providing different types of surrogate models with a surrogate benchmark. Also, in order to guard against a possible case of "bias" in a surrogate benchmark (in the sense of making more accurate predictions for architectures explored by a particular type of NAS optimizer), we recommend to provide two versions of a surrogate: one based on all available training architectures (including those found by NAS optimizers), and one based only on the data gathered for overall coverage.

- Verification: As a means to verify surrogate models, we stress the importance of leave-one-optimizer-out experiments both for data fit and benchmarking, which simulate the benchmarking of 'unseen' optimizers.

- Since most surrogate benchmarks will continue to grow for some time after their first release, to allow apples-to-apples comparisons, we strongly encourage to only release surrogate benchmarks with a version number.

- In order to allow the evaluation of multi-objective NAS methods, we encourage the logging of as many relevant metrics of the evaluated architectures other than accuracy as possible, including training time, number of parameters, and multiply-adds.

- Alongside a released surrogate benchmark, we strongly encourage to release the training data its surrogate(s) were constructed on, as well as the test data used to validate it.

- In order to facilitate checking hypotheses gained using the surrogate benchmarks in real experiments, the complete source code for training the architectures should be open-sourced alongside the repository, allowing to easily go back and forth between querying the model and gathering new data.

## H  REPRODUCIBILITY STATEMENT

For our reproducibility statement, we use the "NAS Best Practices Checklist" (Lindauer & Hutter, 2020), which was recently released to improve reproducibility in neural architecture search.

1. **Best Practices for Releasing Code**

   For all experiments you report:

   (a) Did you release code for the training pipeline used to evaluate the final architectures? [Yes] Follow link in the footnote of page 2 (Section 1). The training pipeline and hyperparameters used to train the architectures are clearly specified in the corresponding scripts.

   (b) Did you release code for the search space [Yes] The used search spaces are the ones from DARTS (Liu et al., 2019) and FBNet (Wu et al., 2019a).

   (c) Did you release the hyperparameters used for the final evaluation pipeline, as well as random seeds? [Yes] Follow link in the footnote of page 2 (Section 1). We released our code, which includes the seeds and final evaluation pipeline used. The hyperparameters for training the architectures are fixed, while the ones for training the surrogate models are optimized using BOHB (Falkner et al., 2018).

   (d) Did you release code for your NAS method? [Yes] All of our code for benchmarking NAS methods is available by following the same link to the codebase.

   (e) Did you release hyperparameters for your NAS method, as well as random seeds? [Yes] The hyperparameters we used are also available.

2. **Best practices for comparing NAS methods**

   (a) For all NAS methods you compare, did you use exactly the same NAS benchmark, including the same dataset (with the same training-test split), search space and code for training the architectures and hyperparameters for that code? [Yes] Refer to Section 3 for details.

   (b) Did you control for confounding factors (different hardware, versions of DL libraries, different runtimes for the different methods)? [Yes] We trained all the architectures on the same GPU (Nvidia RTX2080 Ti) and used the same versions of the used libraries to run all NAS methods.

(c) Did you run ablation studies? [Yes] We show ablation studies in Section 3.1.8.

(d) Did you use the same evaluation protocol for the methods being compared? [Yes]

(e) Did you compare performance over time? [Yes] We typically compare the anytime performance of black-box NAS optimizers for a specified time budget. See the figures in Section 3 for examples.

(f) Did you compare to random search? [Yes] We did include random search in our experiments in Section 3 and data collection.

(g) Did you perform multiple runs of your experiments and report seeds? [Yes] We run the black-box optimizers on the surrogate benchmark multiple times and average the results.

(h) Did you use tabular or surrogate benchmarks for in-depth evaluations? [Yes] This is in fact the main purpose of our paper: to introduce realistic surrogate NAS benchmarks.

3. **Best practices for reporting important details**

(a) Did you report how you tuned hyperparameters, and what time and resources this required? [Yes] We reported this information in Section 3.1.3.

(b) Did you report the time for the entire end-to-end NAS method (rather than, e.g., only for the search phase)? [Yes] Our results use the end-to-end time.

(c) Did you report all the details of your experimental setup? [Yes] We did include all details of the setup in Section 3.

