# OpenReview forum: "Surrogate NAS Benchmarks: Going Beyond the Limited Search Spaces of Tabular NAS Benchmarks"
_ICLR.cc/2022/Conference — ICLR 2022 Poster_

### Official Review · Reviewer_yTPb · 2021-11-01

**Correctness:** 2
**Technical Novelty And Significance:** 2
**Empirical Novelty And Significance:** 2
**Recommendation:** 3
**Confidence:** 5

**Main Review:**

Overall, the paper is well-written but it does not contain significant contributions.
The two main contributions are the use of a surrogate to create a surrogate benchmark and the creation of an interesting new search space for NAS.

I see little contribution of demonstrating how to create a surrogate benchmark for NAS. This has been done earlier for hyperparameter optimization as pointed out by the authors themselves and NAS is nothing but a specific type of hyperparameter optimization. Additionally, Yan et al. presented with their paper "NAS-Bench-x11 and the Power of Learning Curves" already one way to create surrogate benchmarks for NAS.
While the authors claim that a larger search space might be of huge interest for the community, no evidence for this claim was provided. There is no reason to believe that a larger search space is more interesting than a smaller one if not chosen correctly. Furthermore, the search space considered (among others) is the DARTS search space which is well-known and well-explored by the community. Datasets such as CIFAR-10 and CIFAR-100 are used in many NAS papers and the community is already overfitting to these tasks such that they started looking for more challenging problems. Concluding, I do not see this benchmark to be particularly useful.

The authors don't discuss all the shortcomings of their benchmark. It is important to point out that the most efficient NAS methods do not evaluate an architecture completely. Methods such as DARTS or Hyperband search differently and it seems like that these methods cannot be evaluated on this benchmark. While methods exist that evaluate architectures completely, they have no practical purpose since they are simply too expensive to run.

A comment on societal impact: The authors claim that millions of GPU hours will be saved and carbon emissions will be reduced. This is a bold statement and not supported by any evidence. What we have learned from the very first NAS benchmark is that it led to more and more NAS benchmarks, only spending more GPU hours. If the authors want to take the societal impact section seriously, they should consider the risk that they have potentially wasted energy and hardware for no reason.

**Summary Of The Paper:**

The authors present another benchmark tabular benchmark for neural architecture search. In contrast to most other works, only few architectures are actually trained while the remaining ones are imputed using a regression model.

**Summary Of The Review:**

Overall a well-written paper with no significant novelties. The presented benchmark does not seem to be a useful addition to the vast amount of already existing benchmarks.

---

> ### Author Response · Authors · 2021-11-19
> **Reply to Reviewer yTPb (Part 1/4)**
>
> Thank you for taking the time to review our paper. Below we address your major raised concerns.
>
> **"Overall, the paper is well-written but it does not contain significant contributions. The two main contributions are the use of a surrogate to create a surrogate benchmark and the creation of an interesting new search space for NAS."**
>
> We believe there may be a misunderstanding in the second point: the creation of a new search space is neither something we claim as a contribution nor something we do! In contrast, our motivation was to build our first surrogate benchmarks for currently popular search spaces, in order to demonstrate that they do indeed work in these cases without introducing confounding factors.
> If you thought that we’re trying to create an interesting new search space for NAS, then we understand that you may be disappointed that we do not actually fulfill this, but this was never our intention.
>
> **"I see little contribution of demonstrating how to create a surrogate benchmark for NAS. This has been done earlier for hyperparameter optimization as pointed out by the authors themselves and NAS is nothing but a specific type of hyperparameter optimization."**
>
> You are right when saying that NAS can be framed as a hyperparameter optimization problem. Reviewer pRnV also inquired about this relationship, and we have updated the related work in Appendix A with a more detailed discussion on main differences between HPO surrogate benchmarks and the ones proposed in this paper. We also repeat the relevant response we gave to to reviewer pRnV about some of these differences here:
> “The difference in dimensionality between NAS and HPO makes a qualitative difference: in HPO, a benchmark with few hyperparameters makes a lot of sense, as that is actually the most frequent use case of HPO. But this is not the case for NAS, which typically consists of much higher-dimensional spaces. While it certainly wouldn’t come as a surprise to most ML researchers that one can build a strong surrogate benchmark in a, say, 2-dimensional hyperparameter space, most researchers we spoke to before our work on surrogate NAS benchmarks did not expect it to be possible to model the high-dimensional search space of NAS algorithms well enough to build a good-enough surrogate to be useful. In this work, we gave several existence proofs that this is indeed possible. We also showed for the first time that surrogate benchmarks can have lower error than tabular benchmarks and demonstrated an exemplary use of surrogate NAS benchmarks to drive research hypotheses in a case study; neither of these has been done for surrogate HPO benchmarks. Furthermore, we also study the possibility of using pure random search to generate the architectures to base our surrogate on; this turned out to work very well for NAS surrogate benchmarks but would work very poorly for typical higher-dimensional HPO surrogate benchmarks, because many hyperparameters, when set poorly, completely break performance, whereas poor architectural choices still yielded relevant results that could inform the model building.”
>
> In terms of contributions, it seems that there are different opinions from different readers of our work. Citing reviewer pRnV: “The contribution, novelty, analysis of this work is above the most recent published NAS benchmarks.” In the light of the contributions listed above (showing that, against popular belief, it is possible to construct useful surrogate NAS benchmarks despite the high dimensionality of the search space; showing that these surrogates can have lower errors than tabular benchmarks; creating 2 useful surrogate NAS benchmarks, and showcasing how surrogate benchmarks can be used to drive research hypotheses in a case study), we believe that our work provides a significant contribution to the NAS community. Saying that surrogate benchmarks for HPO and the methodology to create them have been there already, is not a valid critique in our opinion and it belittles the effort and time put in adapting such methodology, with all its nitpicks and different design choices, to NAS. As an analogy, NAS-Bench-201 did not have any novelty as such compared to NAS-Bench-101 in terms of methodology. However, it is still a really valuable contribution to the NAS community since it provided a framework to run one-shot NAS algorithms and evaluations on more than one dataset.

---

> > ### Author Response · Authors · 2021-11-19
> > **Reply to Reviewer yTPb (Part 2/4)**
> >
> > **"Additionally, Yan et al. presented with their paper "NAS-Bench-x11 and the Power of Learning Curves" already one way to create surrogate benchmarks for NAS."**
> >
> > We would like to note (without violating the anonymity of this submission) that NAS-Bench-x11 (accepted to the upcoming NeurIPS, and in an early version to a CVPR workshop) is a direct follow-up work of an early arXiv version of our current paper. Our paper comprehensively lays out and evaluates the methodology for creating surrogate NAS benchmarks, and without us having shown that NAS surrogate benchmarks can actually mimic true benchmarks quite closely in terms of trajectories the results in the NAS-Bench-x11 paper would have no basis. In other words, our paper had to be there first for NAS-Bench-x11 to make sense. Furthermore, we’d like to point out that NAS-Bench-x11 is directly utilizing the learning curves we provide in order to build one of their surrogate benchmarks (NAS-Bench-311).
> >
> > **"While the authors claim that a larger search space might be of huge interest for the community, no evidence for this claim was provided. There is no reason to believe that a larger search space is more interesting than a smaller one if not chosen correctly."**
> >
> > We are slightly confused by what you mean exactly. In our eyes the sole purpose of tabular NAS benchmarks is to prototype and improve NAS algorithms so that they can then be used on practical NAS tasks, which tend to have a large search space (see, e.g., the DARTS space, the FB-Net space, etc). If you mean that in practice it is often possible to already obtain good performance by defining a very small search space around the optimal architecture, then, of course, this is true. But that approach can only tackle the comparably uninteresting case of NAS in a space where we already know the optimal region. This might be useful in the short term for beating a benchmark and being able to say that one did it by using NAS, but in the long term, for the scientific study of NAS as a general method for searching for architectures, and as a method that can discover truly novel architectures, we firmly believe that we need to be able to scale NAS to large search spaces.
> >
> > **"Furthermore, the search space considered (among others) is the DARTS search space which is well-known and well-explored by the community."**
> >
> > In case you missed it, we would like to clarify that we did not only use the DARTS search space but also the FBNet one. Please refer to section 3.2 for more details about this surrogate benchmark.  We again would like to stress that we picked the DARTS search space precisely for the reason that it is the most used search space in NAS research and a vast amount of GPU hours are being spent on it.
> >
> > **"The authors don't discuss all the shortcomings of their benchmark."**
> > We discuss the limitations and the guidelines for using surrogate benchmarks in Section 6 and Appendix G.
> >
> > **"It is important to point out that the most efficient NAS methods do not evaluate an architecture completely. Methods such as DARTS or Hyperband search differently and it seems like that these methods cannot be evaluated on this benchmark. While methods exist that evaluate architectures completely, they have no practical purpose since they are simply too expensive to run."**
> >
> > You are right in stating that methods such as DARTS or Hyperband cannot be run on our surrogate benchmarks directly. However, our surrogate benchmarks *can* be used to *evaluate* the performance of arbitrary NAS algorithms that return an architecture, by simply querying the surrogate with that architecture. One can, for example, monitor the anytime performance of the optimal architecture at every epoch during the search for DARTS, as we do in Appendix F.1 (see Figure 22). The same would also apply to arbitrary other NAS algorithms (such as, e.g., Hyperband or zero-shot proxies), since by their very definition as a NAS algorithm they return architectures. This is important since the search efficiency has drastically improved (down to a few minutes) since DARTS was published and the final evaluation phase is the most expensive part of the experimental pipeline; therefore, replacing the cost for this final evaluation is still an important contribution to the entire field of NAS, not only for the field of blackbox methods. Finally, we respectfully disagree with the statement that blacḱ-box NAS methods do not have any practical purpose. Methods such as NAS-BOWL [1] do indeed provide practical insights, e.g. by making the search process and the found architectures more interpretable, and they do indeed report lower search costs and better performance compared to DARTS and many other one-shot NAS algorithms.
> >
> > Finally, we would like to note that NAS-Bench-X11, as a direct follow-up work of the arXiv version of this submission, also allows actually running multi-fidelity methods such as Hyperband and BOHB directly on the benchmarks.

---

> > > ### Author Response · Authors · 2021-11-19
> > > **Reply to Reviewer yTPb (Part 3/4)**
> > >
> > > **"A comment on societal impact: The authors claim that millions of GPU hours will be saved and carbon emissions will be reduced. This is a bold statement and not supported by any evidence. What we have learned from the very first NAS benchmark is that it led to more and more NAS benchmarks, only spending more GPU hours. If the authors want to take the societal impact section seriously, they should consider the risk that they have potentially wasted energy and hardware for no reason."**
> > >
> > > We respectfully, but very firmly, disagree with this point and we are very sorry to hear this perspective; it does appear to reflect a serious misunderstanding of the field of tabular / surrogate NAS benchmarks. The first NAS benchmarks emerged as a resolution to the reproducibility issues existing in NAS research at that time and we believe that they are a tremendous contribution to the NAS community by providing a test-bed that allows benchmarking NAS algorithms without confounding factors (such as different library versions, different hardware, etc.), as well as other disparities such as different data preprocessing or training pipelines. Finally, they also democratized NAS research by making it possible to run NAS algorithms on a single CPU machine, which is important not only for research, but also in teaching NAS to students.
> > >
> > > To put the amount of compute we spent (roughly 60k GPU hours) into perspective: it is only roughly 10% of what Zoph & Le spent for their ICLR 2017 paper on NAS by RL (which used 800 GPUs for 4 weeks). It is also less than 1% of the follow-up paper by Real et al, which carried out 5 runs of each of RL, random search, and regularized evolution. All of these numbers even only take into account the final runs that ended up in the paper, ignoring the compute cost necessary for algorithm development and debugging. Had a surrogate benchmark existed at that time, *none* of the compute in those papers would’ve needed to be spent. Taking into account that the cost of a tabular NAS benchmark can easily be amortized in a single paper, and also taking into account the 200+ citations of each of NAS-Bench-101 and NAS-Bench-201, should demonstrate the clear environmental benefit of creating these benchmarks. By now there are indeed quite a few tabular NAS benchmarks (cited in our paper), which we see as a validation of the concept. However, *none* of these addresses large search spaces, since none of them *can* address large search spaces, for the simple reason that large search spaces cannot be exhaustively evaluated to build a tabular benchmark. It is this fundamental issue that our work resolves. As a result, since our paper has first appeared on arXiv a bit over a year ago, it has already been cited 38 times, with many of the citing papers using our surrogate benchmark to substitute computation that otherwise would’ve had to run on the true DARTS space. As an example, the compute time the authors of the CATE [2] paper would have spent to obtain the results in their Figure 3 (right), if they were to run on the real DARTS benchmark, is roughly 200k GPU hours, which is already more than 3 times the amount of what we spent to create SNB-DARTS. Therefore, the compute cost invested into this paper has already far more than amortized, even before its acceptance.
> > >
> > > Concerning the implications of NAS benchmarks on the environment, we would also like to refer you to the very recent paper [“Towards Green Automated Machine Learning: Status Quo and Future Directions”](https://arxiv.org/abs/2111.05850), which discusses tabular and surrogate NAS benchmarks and concludes by embracing them:
> > >
> > > “While the costs for creating such a benchmark are obviously quite substantial, they represent a one-time investment.
> > > [...]
> > > All in all, [tabular/surrogate] benchmarks are powerful tools to make research on AutoML more sustainable. In particular, they can avoid repetitive evaluations of candidate solutions. This not only helps to save energy, but also enables institutions that cannot afford the necessary resources to research on this topic. Moreover, in general, research can also be accelerated, since evaluations of candidate solutions require only milliseconds instead of minutes, hours or even days. Consequently, the use of benchmarks should clearly be advocated and also requested, since several advantages that benchmarks bring with them can be combined in this
> > > way.”
> > >
> > > We hope to have clarified the misunderstandings of tabular/surrogate NAS benchmarks in general and our paper in particular.
> > >
> > > In general, we do believe that we have clarified all of your concerns and since many of them appear to be based on misunderstandings, we would very much appreciate it if you considered substantially increasing your score.

---

> > > > ### Author Response · Authors · 2021-11-19
> > > > **Reply to Reviewer yTPb (Part 4/4)**
> > > >
> > > > **-- References --**
> > > >
> > > > [1] [Ru et al. 2021. Interpretable Neural Architecture Search via Bayesian Optimisation with Weisfeiler-Lehman Kernels. In ICLR 2021](https://arxiv.org/abs/2006.07556)
> > > >
> > > >
> > > > [2] [Yan et al. CATE: Computation-aware Neural Architecture Encoding with Transformers. In ICML 2021](https://arxiv.org/pdf/2102.07108.pdf)

---

> > > > > ### Comment · Reviewer_yTPb · 2021-11-21
> > > > > **Reply**
> > > > >
> > > > > ## Clarifications
> > > > > I guess my wording was a bit poor regarding the "larger search space". What I meant to say is a "an exhaustively evaluated benchmark" instead of "larger search space". The authors claim that there is a need for that but we simply don't have any evidence for that. To be fair, this is only a minor point.
> > > > >
> > > > > I did not miss that the authors use FBNet as well, that's why it says "among others" in my review.
> > > > >
> > > > > There is a big difference between usage of data and saving computational budget. A benchmark is very useful since it will allow more researchers to work on NAS. However, not every researcher who has used the benchmark manifests into saved energy. Without the benchmark the researcher might not have been able to work on NAS at all. I don't think the authors' post-hoc analysis makes any sense, the truth is that we won't know for sure.
> > > > >
> > > > > ## Main Part
> > > > >
> > > > > I am happy to increase my main points here but my main criticism remains unaddressed. I explicitly reiterate and detail my concerns to give the authors the opportunity to address it directly:
> > > > > * The contribution is not significant since this idea already exists and in fact there is another work which does exactly the same but is better in all aspects.
> > > > > ** I think the authors agree that the idea itself is not novel since it has been done for hyperparameter optimization already. They claim that it is unexpected that it works for NAS problems. Yet, they cite different works, e.g. "Neural Predictor for Neural Architecture Search", which in fact learn from few architectures to successively predict the accuracy of other architectures. In this work, more data is used and to no surprise it works well. A comparison to previous neural architecture prediction or similar surrogate models is missing.
> > > > > ** The authors claim that NAS-Bench-x11 is their own work which they have already published. NAS-Bench-x11 might be a follow-up work, however, it seems one could simply ignore this ICLR submission and only read the NAS-Bench-x11 paper since it is self-contained. What is the interesting aspect we learn from this submission?

---

> > > > > > ### Author Response · Authors · 2021-11-21
> > > > > > **Reply to Reviewer yTPb (Part 1/2)**
> > > > > >
> > > > > > Thank you for your reply and for reiterating again your remaining concerns.
> > > > > >
> > > > > > **"I guess my wording was a bit poor regarding the "larger search space". What I meant to say is a "an exhaustively evaluated benchmark" instead of "larger search space". The authors claim that there is a need for that but we simply don't have any evidence for that."**
> > > > > >
> > > > > > We are a bit confused about what you mean. If we would replace *“larger search space”* with *“an exhaustively evaluated benchmark”* in your original review the sentence would be: *“While the authors claim that an exhaustively evaluated benchmark might be of huge interest for the community, no evidence for this claim was provided. There is no reason to believe that an exhaustively evaluated benchmark is more interesting than a smaller one if not chosen correctly.”*, which does not appear to make sense. Could you please clarify?
> > > > > >
> > > > > > **"There is a big difference between usage of data and saving computational budget. A benchmark is very useful since it will allow more researchers to work on NAS. However, not every researcher who has used the benchmark manifests into saved energy. Without the benchmark the researcher might not have been able to work on NAS at all. I don't think the authors' post-hoc analysis makes any sense, the truth is that we won't know for sure."**
> > > > > >
> > > > > > We agree that we do not exactly know how much energy people would’ve used for NAS experiments without the presence of NAS benchmarks. We only know that *before* their availability, the single paper by [Real et al.](https://arxiv.org/abs/1802.01548), introducing regularized evolution, used 100x the compute that the construction of our entire benchmark required, and that the analyses in that paper could’ve been done for free with our benchmark. Without the availability of NAS benchmarks, the authors of the CATE paper might have chosen to use 200k GPU hours for their experiments (if they have the compute) but they used our benchmark instead, saving that energy. But they might indeed also have chosen to do less evaluations on the true benchmark, which would yield less statistically significant results. What we mean in our argument is that for the same level of experimental rigour the amortized time savings by using a NAS benchmark are large. As such, NAS benchmarks mark a very important milestone for reproducible NAS research (poor reproducibility is something the NAS community was indeed criticized for some years ago) and open-access scientific research.
> > > > > >
> > > > > > **"I think the authors agree that the idea itself is not novel since it has been done for hyperparameter optimization already. They claim that it is unexpected that it works for NAS problems."**
> > > > > >
> > > > > > We again highlight below novel aspects of our work compared to existing surrogate benchmarks in HPO:
> > > > > > - We show that surrogate models can provide a more reliable alternative than tabular entries with a small number of evaluations in terms of both predictive performance and noise modelling (Section 2).
> > > > > > - HPO spaces are typically less complex and constructing a surrogate benchmark on those does not require employing specialized surrogate models such as Graph Neural Networks which operate directly on the graph representation of the neural network architecture.
> > > > > > - We use deep ensembles of the surrogate models in order to model the uncertainty coming from different evaluations and providing a better predictive performance.
> > > > > > - In Section 4 we provide a clear example on how surrogate NAS benchmarks can drive NAS research by offering hints that contradict and help correct conclusions drawn by previous findings that were only true for short runtimes.
> > > > > >
> > > > > > Nevertheless, even though we disagree with you about the novelty of our methods, there are additional dimensions of novelty to take into consideration. There are even many papers that are accepted to venues such as ICLR that only contain empirical studies [[1](https://arxiv.org/abs/1912.12522), [2](https://arxiv.org/pdf/1902.08142.pdf), [3](https://arxiv.org/abs/2104.01177), [4](https://arxiv.org/abs/1906.02530), [5](https://arxiv.org/pdf/2007.01547.pdf)]. We think that this line of work is very useful to any community in general. In our opinion, there is quite some difference between having a hunch on the fact that surrogate NAS benchmarks can be constructed and actually going through all the details in constructing and properly evaluating them.

---

> > > > > > > ### Author Response · Authors · 2021-11-21
> > > > > > > **Reply to Reviewer yTPb (Part 2/2)**
> > > > > > >
> > > > > > > **"Yet, they cite different works, e.g. "Neural Predictor for Neural Architecture Search", which in fact learn from few architectures to successively predict the accuracy of other architectures. In this work, more data is used and to no surprise it works well. A comparison to previous neural architecture prediction or similar surrogate models is missing."**
> > > > > > >
> > > > > > > Note that in Section 2 we show that predictions coming from a surrogate model can (surprisingly) be a better alternative to tabular entries by yielding a stronger predictive performance and better noise modelling, even when only a subset of the training data is available. Moreover, we did indeed compare many surrogate models (see Section 3.1.1 and Table 2) and provided a descriptive comparison of them in Appendix A.2. We would like to emphasize that the main scope of our work is to provide a detailed methodology on how to construct surrogate benchmarks for NAS. We do not claim anywhere that the surrogate models we used in our paper are the best, nor that we introduce a new performance predictor. Stronger performance predictor in the future would only make our case on constructing and using surrogate NAS benchmarks stronger, since we can very easily improve the existing surrogate benchmarks by just fitting them on the existing data that was already collected. Also note that there is a crucial difference in using surrogate models inside a NAS algorithm (which is the main goal in the paper you mentioned above) and in using them to construct surrogate benchmarks. Please refer to the last paragraph in Section 3.1.1 where we discuss this in more details.
> > > > > > >
> > > > > > > **"The authors claim that NAS-Bench-x11 is their own work which they have already published. NAS-Bench-x11 might be a follow-up work, however, it seems one could simply ignore this ICLR submission and only read the NAS-Bench-x11 paper since it is self-contained. What is the interesting aspect we learn from this submission?"**
> > > > > > >
> > > > > > > We would really appreciate if the reviewer would point out where we claimed that NAS-Bench-x11 is our own work. We never claimed that; we said that it is follow-up work drawing on this current paper. Note that NAS-Bench-x11 is a direct extension of our methodology for constructing surrogate benchmarks,  tailored to learning curves, and it clearly cites this current paper as prior work. NAS-Bench-x11 never evaluates NAS methods on both the real and surrogate benchmarks to verify that the trajectories obtained are similar, but, based on this current paper, takes it as a given that surrogate NAS benchmarks work. Furthermore, the fact that surrogate NAS benchmarks can be a better alternative to the tabular ones (which we demonstrate in motivational Section 2) is also not investigated in the NAS-Bench-x11 paper. Other aspects which we think might be of particular interest, are the fact that we show that surrogate NAS benchmarks can be built only utilizing randomly sampled architectures and the fact that we demonstrate that surrogate benchmarks can drive NAS research and lead to correct conclusions which would otherwise be different if NAS algorithms were to be run for a shorter time.  would also like to note that, even if the contributions of the NAS-Bench-x11 paper subsumed ours (which, as just argued, is not the case), the NAS-Bench-x11 has only publically appeared on arXiv on November 5th, 2021, i.e., long after the ICLR submission deadline, and should thus definitely not be seen as prior art that we need to distinguish ourselves from (especially seeing that that work cites ours as prior art). We do understand the reviewer‘s confusion due to the rare publication of a speedy, independent, follow-up work before the publication of the original work, but we hope our clarifications help you to understand the situation and to adjust your assessment of our work. Thank you for your service to ICLR!

---

> > > > > > > > ### Comment · Reviewer_yTPb · 2021-11-24
> > > > > > > > **Answer**
> > > > > > > >
> > > > > > > > **"We would really appreciate if the reviewer would point out where we claimed that NAS-Bench-x11 is our own work."**
> > > > > > > >
> > > > > > > > Sorry, for jumping to conclusions. I read *"We would like to note (without violating the anonymity of this submission) that NAS-Bench-x11 (accepted to the upcoming NeurIPS, and in an early version to a CVPR workshop) is a direct follow-up work of an early arXiv version of our current paper."* and assumed that you meant that this is your own prior work. I was probably understanding this because I read both papers before and I know that the joint set of authors is not empty. Therefore, I am a bit confused that you are now claiming that your work is independent.
> > > > > > > >
> > > > > > > > Disclaimer: No idea whether reviewers should pretend as if they wouldn't know about that fact. But since the papers share commonalities, I think it should be considered.
> > > > > > > >
> > > > > > > > **"We discuss the limitations and the guidelines for using surrogate benchmarks in Section 6 and Appendix G."**
> > > > > > > >
> > > > > > > > I was not able to find any comment on the limitation I was speaking about, i.e. that it cannot be used to evaluate Hyperband or similar methods at arbitrary budgets. I agree with you that NAS-Bench-X11 is not having this limitation which is exactly my point.
> > > > > > > >
> > > > > > > > **"Yet, they cite different works, e.g. "Neural Predictor for Neural Architecture Search", which in fact learn from few architectures to successively predict the accuracy of other architectures. In this work, more data is used and to no surprise it works well"**
> > > > > > > >
> > > > > > > > This was a comment on *"While it certainly wouldn’t come as a surprise to most ML researchers that one can build a strong surrogate benchmark in a, say, 2-dimensional hyperparameter space, most researchers we spoke to before our work on surrogate NAS benchmarks did not expect it to be possible to model the high-dimensional search space of NAS algorithms well enough to build a good-enough surrogate to be useful. In this work, we gave several existence proofs that this is indeed possible."* to point out that this shouldn't be too surprising to the community.
> > > > > > > >
> > > > > > > > **"Furthermore, the fact that surrogate NAS benchmarks can be a better alternative to the tabular ones (which we demonstrate in motivational Section 2) is also not investigated in the NAS-Bench-x11 paper."**
> > > > > > > >
> > > > > > > > Isn't that what they show in Table 4-6?
> > > > > > > >
> > > > > > > > **"NAS-Bench-x11 has only publically appeared on arXiv on November 5th, 2021, i.e., long after the ICLR submission deadline, and should thus definitely not be seen as prior art that we need to distinguish ourselves from"**
> > > > > > > >
> > > > > > > > Wasn't it accessible since May on openreview? Given that these two papers share authors, I think it is fair to make an exception to this general rule.

---

> > > > > > > > > ### Author Response · Authors · 2021-11-25
> > > > > > > > > **Reply to Reviewer yTPb (Part 1/2)**
> > > > > > > > >
> > > > > > > > > **"Therefore, I am a bit confused that you are now claiming that your work is independent."**
> > > > > > > > >
> > > > > > > > > By  “independent”, we meant temporally and spatially separate. Our work was first, and, after it was publicly available on arXiv the NAS-Bench-x11 project team formed and simply used that publicly available version of our paper, as well as our publicly available data, as a basis for their separate follow-up work. No additional information flowed through the single overlapping author, since the current work was essentially done by that time and didn’t change anymore. Also, no information flowed back; the main change to the current paper from previous versions was the addition of a surrogate on the FBNet space, something that was not done in NAS-Bench-x11. (We note that this change was crucial, though, to show another successful surrogate next to the one on the DARTS space.)
> > > > > > > > >
> > > > > > > > > Since the NeurIPS paper on NAS-Bench-x11 was not publicly available by the ICLR submission deadline we could not cite it as a follow-up. However, now that it is available, we are happy to add the following paragraph to our related work in order to clarify the relationship between the two papers and to also address the limitation to modelling final performance: *“A limitation of the surrogate benchmarks we construct in this paper is the fact that they provide only single predictions at the final training epoch for every architecture. NAS-Bench-x11 [Yan et al. 2021], a direct follow-up work of this paper, extends our methodology to predicting full learning curves. This also allows benchmarking multi-fidelity black-box optimizers, such as HyperBand [Li et al. 2017] or BOHB [Falkner et al. 2018], which is not possible in SNB-DARTS or SNB-FBNet. NAS-Bench-x11 extends SNB-DARTS by utilizing the learning curve information of the 60k architectures that we have already released. The main contribution of NAS-Bench-x11 is the surrogate model tailored to predict the full learning curve information, while their methodology to create surrogate NAS benchmarks is based on ours. We emphasize that the possibility to fit surrogate NAS benchmarks to model entire learning curves further strengthens the line of work on surrogate NAS benchmarks we propose.”*. We will also add a note about this in the “Guidelines & Limitations” of Section 6 and Appendix G.
> > > > > > > > >
> > > > > > > > > A further difference to the NAS-Bench-x11 paper is that, that paper never evaluates NAS methods on both the real and surrogate benchmarks to verify that the trajectories obtained are similar, but, based on this current paper, takes it as a given that surrogate NAS benchmarks work. The fact that surrogate NAS benchmarks can be a better alternative to tabular ones (which we demonstrate in the motivational Section 2) is also a key contribution of our current paper, as is the case study for using NAS benchmarks to drive scientific hypotheses .*
> > > > > > > > >
> > > > > > > > > **""Yet, they cite different works, e.g. "Neural Predictor for Neural Architecture Search", which in fact learn from few architectures to successively predict the accuracy of other architectures. In this work, more data is used and to no surprise it works well". This was a comment on "While it certainly wouldn’t come as a surprise to most ML researchers that one can build a strong surrogate benchmark in a, say, 2-dimensional hyperparameter space, most researchers we spoke to before our work on surrogate NAS benchmarks did not expect it to be possible to model the high-dimensional search space of NAS algorithms well enough to build a good-enough surrogate to be useful. In this work, we gave several existence proofs that this is indeed possible." to point out that this shouldn't be too surprising to the community."**
> > > > > > > > >
> > > > > > > > > We would like to emphasize that our quoted response discussed strong  *surrogate benchmarks*. In the "Neural Predictor for Neural Architecture Search" paper the performance predictor is used to find good performing architectures and the focus is only on that. For instance, the samples in Figure 10 (right) in that paper are selected as the 10 best architectures based on their predicted performance and then retrained from scratch in order to assess the quality of the prediction. In contrast, when creating surrogate *benchmarks*, we want a good global coverage of the space which would yield reliable simulated learning curves for various NAS optimizers. To the best of our knowledge, there is no previous work that uses performance predictors in order to create NAS surrogate benchmarks.

---

> > > > > > > > > > ### Author Response · Authors · 2021-11-25
> > > > > > > > > > **Reply to Reviewer yTPb (Part 2/2)**
> > > > > > > > > >
> > > > > > > > > > **"Isn't that what they show in Table 4-6?"**
> > > > > > > > > >
> > > > > > > > > > This is indeed an experiment that shows a comparison of the surrogate benchmark to a tabular version. However, it is not claimed as a novelty in the NAS-Bench-x11 paper and we will coordinate with its authors to cite our paper again in that appendix to avoid any confusion in the reader as to which paper showed better performance than a tabular benchmark first. (We’d like to note, though, that generally, the NAS-Bench-x11 paper very clearly cites us as prior work.)
> > > > > > > > > >
> > > > > > > > > > **"Wasn't it accessible since May on openreview?"**
> > > > > > > > > >
> > > > > > > > > > No, NeurIPS submissions were not publicly accessible during the review period.
> > > > > > > > > >
> > > > > > > > > >
> > > > > > > > > > **-- References --**
> > > > > > > > > >
> > > > > > > > > > Yan et al. *NAS-Bench-x11 and the Power of Learning Curves*. In NeurIPS 2021
> > > > > > > > > >
> > > > > > > > > > Li et al. *Hyperband: A Novel Bandit-Based Approach to Hyperparameter Optimization*. In ICLR 2017
> > > > > > > > > >
> > > > > > > > > > Falkner et al. *BOHB: Robust and Efficient Hyperparameter Optimization at Scale*. In ICML 2018

---

### Official Review · Reviewer_1eb8 · 2021-11-02

**Correctness:** 4
**Technical Novelty And Significance:** 4
**Empirical Novelty And Significance:** 4
**Recommendation:** 8
**Confidence:** 4

**Main Review:**

Pros:
- **Clear motivation**: This work addresses 1) the limitations of small tabular benchmarks, 2) the expensive time and compute cost of large non-tabular search space. For instance, 60k models are sampled from the DARTS search space and their performance are predicted using the surrogate model. This creates a tabular benchmark which is much bigger than NAS-Bench-2.
- **Comprehensive analysis**: The paper has considered potential issues of using surrogate models and has proposed ways to work around them. For example, to yield good coverage of the search space and to train a strong surrogate model, a sampling scheme is used to collect models by random search as well as high performing models by 10 NAS methods.
- **Guidline for future research**: The paper summarizes several best practices to create surrogate NAS benchmarks for new search spaces. It is particular important as there are many promising architectures that are not currently covered in DARTS, or NAS-Bench-1/2. For example, research in transformer-like architectures are increasingly popular. This paper provides a feasible method to fast evaluate new search spaces.

Cons:
- **Search trajectories**: The paper has done lots of experiments to show that the search trajectories running on the surrogate benchmark closely resemble the ground truth. However, it is unclear if the searched models (via surrogate and via ground truth) are also similar. For example, a surrogate may underestimate the error of certain region of the search space, and the NAS method may be guided to favour models in that region. Even though the search trajectory may look right, the models explored could be different from the ground truth. If you take the models searched by different NAS methods on SNB-DARTS and train them, do they still preserve the same ranking?
- **Performance on unseen models**: In SNB-DARTS, 60k architectures are sampled and 0.8 of them are used to train the surrogate model. Since the search is performed on the same 60k search space, it is not suprising for the surrogate model to perform well. How about the unseen models in the DARTS search space? Similarly, NAS-Bench-1 has 423k architectures, which is bigger than SNB-DARTS. Any insight on how SNB-NAS-Bench-1 compare to the tabular NAS-Bench-1?

**Summary Of The Paper:**

The authors propose a method to create surrogate benchmark for NAS. By modelling and predicting the performance of neural architecture in the search space, a much larget search space can be covered without expensive training.

**Summary Of The Review:**

Overall, the paper is very well written. It is clearly motivated and supported by lots of ablation studies. The dataset and search space might not be SOTA, but it does provide a promising practice for future NAS research. There are a few questions that I would like the authors to clarify, otherwise, I would recommend acceptance of this paper.

---

> ### Author Response · Authors · 2021-11-19
> **Thank you, we have added the suggestions to the paper.**
>
> Thank you very much for your feedback and for recommending acceptance of our paper. Below we address all your concerns.
>
> **"If you take the models searched by different NAS methods on SNB-DARTS and train them, do they still preserve the same ranking?"**
>
> This is a very valid point and we had planned to conduct such an experiment too. We took the trajectory of regularized evolution (RE) on the SNB-FBNet (since there we did less function evaluation than in SNB-DARTS, therefore it was affordable to compute during this rebuttal period) and retrained all the architectures from scratch. We show the results in Figure 28 in the Appendix, where in the x-axis we plot the validation error of all the datapoint in one RE run on SNB-FBNet (one of the red line runs in Figure 5, right plot) vs. the validation error values of the same configurations when retrained from scratch in the y-axis. As expected the rank correlation is the same as the one we have already reported on the held-out test set of the randomly sampled architectures, i.e. a Kendall Tau correlation coefficient of 0.73 and Spearman correlation coefficient of 0.895). We expect that this should be the same for SNB-DARTS, i.e. based from Table 2, a Kendall Tau correlation coefficient of more than 0.8. For completeness, we will add a similar plot as the one in Figure 28 for SNB-DARTS for the camera ready version of the paper.
>
> **"Performance on unseen models: In SNB-DARTS, 60k architectures are sampled and 0.8 of them are used to train the surrogate model. Since the search is performed on the same 60k search space, it is not suprising for the surrogate model to perform well. How about the unseen models in the DARTS search space? Similarly, NAS-Bench-1 has 423k architectures, which is bigger than SNB-DARTS. Any insight on how SNB-NAS-Bench-1 compare to the tabular NAS-Bench-1?"**
>
> We believe there is a misunderstanding here. The search on the SNB-DARTS space is not carried out on the subset of 60k evaluated architectures, but on the entire search space with 10^18 architectures in there. Therefore, (10^18-60k) / 10^18 = 99.999999999994% of the SNB-DARTS search space is indeed unseen at training time. There is an extremely small chance (0.000000000006%) that a random architecture a NAS optimizer samples has actually been used for training the surrogate, but we do not expect this to affect our results. Furthermore, for all our experiments regarding model performance, we partitioned our 60k evaluated architectures into disjoint training, validation, and test splits.
>
> We are somewhat confused by your statement *“Similarly, NAS-Bench-1 has 423k architectures, which is bigger than SNB-DARTS.”*. It is true that the *number of evaluated architectures* is larger for NAS-Bench-101 than the number (60k) of evaluated architectures in SNB-DARTS, but the actual search space is dramatically larger for SNB-DARTS (namely, 10^18 architectures). For NAS-Bench-101, since the total number of architectures in the space is much smaller than in SNB-DARTS, it is true that the NAS optimizers might sample an architecture that is contained in the training set of the surrogate model. However, even in this case, the comparison to the tabular case remains fair, because for the tabular case, *all* architectures were in the training set. We mainly used NAS-Bench-101 and the surrogate models fitted on subsets of it to motivate the benefits of surrogate benchmarks. Nevertheless, we now added Appendix F.4 and a plot (Figure 27) that shows the true trajectories of RS and RE vs. their trajectories when run on the surrogate counterpart of NAS-Bench-101.
>
> We hope to have clarified all of your concerns. Please do not hesitate to post if you have any further questions or concerns. If we clarified everything, we would very much appreciate if you would consider increasing your score.

---

> > ### Comment · Reviewer_1eb8 · 2021-11-20
> > **Thank you for the clarification**
> >
> > Thank you for the extra experiment on SNB-FBNet showing the rank correlation of the search trajectory. This addressed my concerns. Please add the results for SNB-DARTS as well.
> >
> > The part about unseen models in the SNB-DARTS space is also clear to me now. There was a confusion about the data being used to train the surrogate model and that used to search. I don't have any more questions.
> >
> > Overall, this paper looks solid to me. I have increased my score to support the acceptance.

---

> > > ### Author Response · Authors · 2021-11-21
> > > **Thank you for increasing your score**
> > >
> > > We are happy to see that our reply addressed your concerns. Thank you very much for increasing your score!

---

### Official Review · Reviewer_pRnV · 2021-11-02

**Correctness:** 4
**Technical Novelty And Significance:** 4
**Empirical Novelty And Significance:** 4
**Recommendation:** 8
**Confidence:** 5

**Main Review:**

In general, this is a good NAS paper that explored a new direction -- surrogate NAS benchmarks. I believe this work stand between 8 and 10. Please see my detailed comments below.

Strengths:
- In this work, the authors proposed a novel family of NAS benchmark -- surrogate NAS benchmark. This can break the limitations of existing NAS benchmarks -- the search space is unrealistically small.
- The effectiveness and usability of the proposed new surrogate NAS benchmarks have been systemically evaluated and analyzed.
- This work brings some fresh ideas and artifacts to the NAS community. Given various bottlenecks, the development of the NAS research is significantly slowed down. This work can potentially help bring some new insights.
- Section 2 is interesting that the surrogate model can yield strong/better predictive performance than standalone training.
- All codes have been released.
- We can see many NAS benchmarks were accepted in the top venues this year. The contribution, novelty, analysis of this work is above the most recent published NAS benchmarks.

Weakness:
- As surrogate HPO benchmarks have been proposed, it would be good to have a deep analysis on the comparison with that.
- I do not find other clear weaknesses.

Minor issues:
- In the first paragraph, the authors cited (Hao, 2019) for carbon emissions. A more proper reference for the carbon emissions of NAS algorithms would be "Carbon Emissions and Large Neural Network Training" and "Full-Cycle Energy Consumption Benchmark for Low-Carbon Computer Vision", which particularly discussed the carbon emissions of NAS methods.
- Some works have been officially accepted and then it is suggested to cite their official version instead of arxiv version, such as Dong et al 2020 NATS-Bench arxiv 2020 -> TPAMI 2021

**Summary Of The Paper:**

This work explored how to use surrogate models to expand the existing (and limited) neural architecture search -- NAS -- benchmark. The new expanded benchmark is named as surrogate NAS benchmark. All codes are open-sourced, which demonstrated the good reproducibility of this work. The authors have conducted extensive experiments to demonstrate the usability of this new surrogate NAS benchmark and showed much analysis of existing NAS methods on this new benchmark.

**Summary Of The Review:**

A good paper for the NAS community; hits a sweet spot for the current bottleneck of the reproducible NAS and the scientific research of NAS. I would strongly recommend accepting this work for ICLR.

---

> ### Author Response · Authors · 2021-11-19
> **Thank you for the positive feedback and acceptance score**
>
> Many thanks for your positive feedback and the acceptance score. We are really happy to read that you have a high opinion about our work and that you said that “the contribution, novelty, and analysis of this work is above the most recent published NAS benchmarks.”. We updated the paper accordingly with the minor issues you found, thanks! Below we discuss your main concern:
>
> **"As surrogate HPO benchmarks have been proposed, it would be good to have a deep analysis on the comparison with that."**
>
> Thank you for this comment, we agree that it is useful to highlight the similarities and differences of surrogate benchmarks for NAS and HPO and have updated the related work in Appendix A accordingly. Ultimately, the NAS problem can indeed be formulated as an HPO problem and some algorithms designed for HPO can indeed be evaluated on NAS benchmarks; however, this does require that they can handle the corresponding high-dimensional categorical hyperparameter space typical of NAS benchmarks (e.g., 34 hyperparameters for the DARTS space; 22 hyperparameters for the FBNet space). This difference in dimensionality makes a qualitative difference: in HPO, a benchmark with few hyperparameters makes a lot of sense, as that is actually the most frequent use case of HPO. But this is not the case for NAS, which typically consists of much higher-dimensional spaces. While it certainly wouldn’t come as a surprise to most ML researchers that one can build a strong surrogate benchmark in a, say, 2-dimensional hyperparameter space, most researchers we spoke to before our work on surrogate NAS benchmarks did not expect it to be possible to model the high-dimensional search space of NAS algorithms well enough to build a good-enough surrogate to be useful. In this work, we gave several existence proofs that this is indeed possible. We also showed for the first time that surrogate benchmarks can have lower error than tabular benchmarks and demonstrated an exemplary use of surrogate NAS benchmarks to drive research hypotheses in a case study; neither of these has been done for surrogate HPO benchmarks. Furthermore, we also study the possibility of using pure random search to generate the architectures to base our surrogate on; this turned out to work very well for NAS surrogate benchmarks but would work very poorly for typical higher-dimensional HPO surrogate benchmarks, because many hyperparameters, when set poorly, completely break performance, whereas poor architectural choices still yielded relevant results that could inform the model building. Finally, NAS benchmarks are also far more expensive than typical HPO benchmarks. This calls even more for the necessity of a careful methodology on how to construct surrogate benchmarks for NAS.

---

> > ### Comment · Reviewer_pRnV · 2021-11-21
> > **Reasonable Explanation & Keep the Rating of Strong Accept.**
> >
> > The authors' responses make a lot of sense to me. The new deep analysis on the relationship between surrogate HPO and surrogate NAS addressed my concerns.
> > All my concerns have been addressed in the new revision and I keep my original rating "Strong Accept" for this work.

---

### Official Review · Reviewer_t8ZB · 2021-11-03

**Correctness:** 3
**Technical Novelty And Significance:** 2
**Empirical Novelty And Significance:** 2
**Recommendation:** 5
**Confidence:** 4

**Main Review:**

Pros.
1. the authors have made several significant improvement over the past submission and I'm happy to see that. Compared to the prior submission, the authors have included results on CIFAR10, CIFAR100, and ImageNet. I thank authors for doing that, however, there is still some concerns for me to see this work get published.

Cons:
1. Apparently this work is using black box optimization to perform NAS. However the selected benchmark baselines are pretty old. I'd suggest authors taking a look at the recent public benchmark results from here: https://bbochallenge.com/ (the black box optimization challenges held at NeurIPS 2021). And update the paper accordingly to track the recent advance in the field. Essentially NAS is no different from those test functions.
2. All the results are plotted by the wall time, which is a bit tricky in NAS. The costs of querying a surrogate model can be cheap, but sample-efficiency (#samples over accuracy) is far more important to NAS as the cost of evaluations are the main bottleneck, not the search. Please correct me if I'm wrong, but I don't think you consider the cost of evaluations in the wall time.
3. Please diversify the tasks. Here I mean different tasks, e.g. detection or segmentation, rather than image recognition with different datasets and models.





**Summary Of The Paper:**

Overview: This work proposes a new NAS benchmark based on the results of surrogate models prediction. This surrogate model is able to predict all architectures in DARTS search space, which is about 10^18 possible architectures. The author compared the predict performance among different type of surrogate models and also leveraged surrogate models to investigate different NAS methods.

**Summary Of The Review:**

I have been reviewing this paper for a few times, and I appreciate the effort that the authors have put to update the paper. However, the current draft can not truly reflect the recent advancement in the field, and it may mislead the future. Once the author have address my concerns 1, 2, I'm open to accept this paper.

---

> ### Author Response · Authors · 2021-11-19
> **Thank you for your review and the useful suggestions**
>
> Thank you for taking their time to review our paper again, and we are happy to know that you consider the additional changes made to the submission significant. We reply to all comments below:
>
> **"Apparently this work is using black box optimization to perform NAS. However the selected benchmark baselines are pretty old. I'd suggest authors taking a look at the recent public benchmark results from here: https://bbochallenge.com/ (the black box optimization challenges held at NeurIPS 2021). And update the paper accordingly to track the recent advance in the field. Essentially NAS is no different from those test functions."**
>
> We agree that the NAS problem is essentially not different from the hyperparameter optimization one. The main reason that initiated the development of NAS as another separate sub-field of AutoML was the complex graph-structured architectural spaces, which are typically conditional and categorical high-dimensional ones. This has led to the development of dedicated algorithms that operate efficiently and effectively on such spaces, while most work on hyperparameter optimization focusses on low-dimensional continuous problems. Nevertheless, the BBO challenge you refer to is indeed one of the few exceptions in which algorithms also need to handle categorical choices. While none of the benchmarks in the competition had remotely as many hyperparameters as there are choices to be made in NAS for the search spaces we’re dealing with (34 for the DARTS search space and 22 for the FBNet search space), the algorithms do syntactically work on these search spaces. We therefore now integrated the winner of this BBO challenge, HEBO [1], in our codebase. It is currently running, which will still take a few days (since we need to also run it on the expensive blackbox itself in order to compare performance on that and on the surrogate); we will provide an update here (and will update Figure 5) before the rebuttal period ends. Thank you for this comment as we agree that this makes the paper stronger.
>
> **"All the results are plotted by the wall time, which is a bit tricky in NAS. The costs of querying a surrogate model can be cheap, but sample-efficiency (#samples over accuracy) is far more important to NAS as the cost of evaluations are the main bottleneck, not the search. Please correct me if I'm wrong, but I don't think you consider the cost of evaluations in the wall time."**
>
> This is actually a misunderstanding. We apologize if this was not clear enough in the paper, but all the plots in the paper that show the NAS optimizers’ trajectories *do* consider these evaluation costs. The x-axis for plots on the true benchmark (like Figure 2a) show the sum of the search costs (time taken by the NAS algorithm to suggest the next architecture) and the true evaluation cost (time to train the proposed architecture by the NAS algorithm), while the x-axis for plots on surrogate benchmarks (like Figures 2b and 2c) show the sum of the search costs and the *estimated* evaluation cost. This estimated evaluation cost utilizes a surrogate model fitted on the runtime in order to predict the evaluation cost (i.e. training time of every architecture); details for this are explained in Section 3.1.4. We hope that this clarifies this misunderstanding, but would be happy to clarify any further questions on it.
>
> **"Please diversify the tasks. Here I mean different tasks, e.g. detection or segmentation, rather than image recognition with different datasets and models."**
>
> Thanks, we agree that the diversity of tasks that we believe our surrogate NAS benchmark methodology enables would best already be exemplified in this paper introducing that methodology. We have therefore started creating a new surrogate benchmark for an RNN space on the PennTreeBank dataset. We are aiming for a median test perplexity value below 70. This takes around 1.5h per architecture to train; therefore, this benchmark will not be complete in the limited rebuttal time, but we would gladly add it to the camera-ready-copy of the paper.
> Thank you again for the very useful feedback. We hope that we addressed all your concerns and that you will consider updating your score as promised after our response.
>
>
> **-- References --**
>
> [1] Cowen-Rivers et al. An Empirical Study of Assumptions in Bayesian Optimisation.

---

> > ### Author Response · Authors · 2021-11-21
> > **We have updated the paper with the HEBO (winner of https://bbochallenge.com/) results. Thank you for the suggestion!**
> >
> > We thank the reviewer for the very useful suggestion. The HEBO experiment finished by now and we included its result in Figure 5. As the results show, HEBO’s competitiveness indeed extends to high-dimensional categorical NAS spaces. We have updated the text in section 3.2 accordingly with the new HEBO results and added a description of HEBO in page 21. Interestingly, HEBO performed much better than RE in terms of anytime performance and slightly better then RE in the end. We hope that with this experiment we have properly addressed your concerns and you would consider increasing your score as you indicated.

---

> > > ### Author Response · Authors · 2021-11-27
> > > **Remaining concerns about points 1 and 2**
> > >
> > > Thank you again for your review. We believe to have fully addressed your concerns 1 and 2 (with the creation of a new benchmark for concern 3 taking longer than the rebuttal period). We were glad to read that you would be open for accepting the paper with concern 1 and 2 taken care of and were wondering whether there you have any remaining concerns about these points, which we would be happy to address. Thanks!

---

### Decision · Program_Chairs · 2022-01-20

**Decision:**

Accept (Poster)

**Comment:**

This paper proposes a methodology to create cheap NAS surrogate benchmarks for arbitrary search spaces. Certainly, the work is interesting and useful, with comprehensive studies to validate such approach. It should be credited as belonging to the first efforts of introducing and comprehensively studying the concept of surrogate NAS benchmarks. In AC's opinion, it is a solid paper that will (or has already) inspire many follow up works. The paper is well written.

This paper received highly mixed ratings. Although the authors might not see, all reviewers actually participated in the private discussions. Reviewer 1eb8 indicated hesitation in her/his support. Reviewer yTPb stated that if not considering the arXiv complicacy, she/he "would certainly raise score by one level".  AC also reached out to Reviewer yTPb about her/his mentioned possibility of updating scores, and got confirmed that her/his original opinions wasn't changing after rebuttals. Besides, AC agrees the arXiv/NeurIPS complicacy shouldn't brought into the current discussion, and ignored that factor during decision making.

The main sticking (and considered-as-valid) critique is on the relatively outdated and incomplete selection of baselines. As a benchmark paper, it should capture and diversify the recent methods. For example, the authors might consider adding: https://botorch.org/docs/papers (latest methods in Bayesian Optimization) https://github.com/facebookresearch/LaMCTS (latest methods in Monte Carlo Tree Search) https://facebookresearch.github.io/nevergrad/ (latest methods in Evolutionary algorithms)

Given the above concerns, AC considers this paper to sit on the borderline, and perhaps with pros outweighing the cons. Hence, a weak accept decision is recommended at this moment.